# Vibronically assisted sub-cycle charge transfer at a non-fullerene acceptor heterojunction

Pratyush Ghosh [1], Jeroen Royakkers[2], Giacomo Londi [3], Samuele Giannini [3], Rakesh Arul [1], Alexander J. Gillett [1,4], Scott T. Keene [1,5], Szymon J. Zelewski [1,6], David Beljonne [7]✉, Hugo Bronstein [1,2]✉ & Akshay Rao [1]✉

Excited-state charge transfer underpins organic photovoltaics, photocatalysis and photodetection, but is traditionally thought to require large energy offsets and strong donor–acceptor coupling that can limit device performance. Here, we investigate through-space polymer non-fullerene-acceptor based model heterojunctions in which a perylene diimide acceptor is covalently tethered to a low-bandgap polymer donor. These systems feature an exceptionally small energy offset (< 100 meV) between frontier orbitals, with weak donor–acceptor coupling in the Franck–Condon region. We nevertheless achieve a charge-transfer timescale of ~18 fs. This ultrafast charge-transfer is accompanied via the launch of coherent wavepackets along a high-frequency vibrational coordinate (26 fs period) on the non-fullerene acceptor's potential energy surface. We uncover specific polymer-centered driving vibrational modes that enable such rapid charge-transfer rates, by mixing Frenkel exciton and charge-transfer states following photoexcitation. Our results demonstrate that ultrafast charge-transfer can be achieved—ultimately limited by high-frequency vibrational periods—even in the absence of large energy offsets or strong ground-state coupling.

Photoexcited charge transfer (CT) plays a pivotal role in organic photovoltaics (OPV)[1–3], photosynthesis[4] and photocatalysis[5–7]. In such light-harvesting applications, dissociation of excitons via electron (or hole) transfer to an electron acceptor (or donor) allows for the eventual generation of free charges, used to generate photocurrent or drive redox chemistry[1,2,4,6,8].

One key design principle for improving the performance of light-harvesting devices is to enhance the CT rate at the heterojunction between the donor and acceptor molecules. This ensures that photo-generated excitons form free charges faster than they radiatively or non-radiatively recombine. A key question is what ultimately controls the rate of CT. Typically for polymer-fullerene-based heterojunctions, large energetic offsets between frontier orbitals (≥500 meV) have enabled ultrafast CT on sub-100 fs timescales[9–11]. However, such large offsets limit the maximum achievable open-circuit voltage in photovoltaics or increase the overpotential in photocatalysis. Recent developments in non-fullerene acceptors[12] (NFAs) have demonstrated that fast (~100–250 fs) CT can occur even with low driving forces[13–15]—

[1]Cavendish Laboratory, University of Cambridge, Cambridge, UK. [2]Yusuf Hamied Department of Chemistry, University of Cambridge, Cambridge, UK. [3]Department of Chemistry and Industrial Chemistry, University of Pisa, Pisa, Italy. [4]Department of Physics, Chemistry and Biology (IFM), Linköping University, Linköping, Sweden. [5]Department of Materials Science and NanoEngineering, Rice University, Houston, TX, USA. [6]Department of Experimental Physics, Wroclaw University of Science and Technology, Wroclaw, Poland. [7]Laboratory for Chemistry of Novel Materials, University of Mons, Mons, Belgium. ✉e-mail: David.BELJONNE@umons.ac.be; hab60@cam.ac.uk; ar525@cam.ac.uk

but this is not universally observed[16]. In many cases, similar systems with low energetic offsets exhibit much slower dynamics, extending from tens of ps to ns[16–20]. This inconsistency points to a gap in understanding which factors govern the ultimate rate of CT when the energetic offset is small, and how we can achieve the fastest possible electron transfer in that regime.

Traditionally, the rate of CT in light-harvesting devices is believed to be primarily governed by the electronic coupling and energy offset between the frontier orbitals between donor and acceptor molecules[21,22]. Recent studies on symmetry-breaking CT (SBCT) in covalent dimers and ordered aggregates have shown that strong electronic coupling can enable ultrafast, low-energy-loss CT[23–25]. However, the role of vibrational modes—whether they assist, limit, or modulate the CT process in weakly coupled organic heterojunctions—remains poorly understood. Understanding this interplay is essential for identifying the fundamental speed limits of photoinduced charge separation in molecular heterojunction-based photovoltaic devices.

Here, we show that CT in weakly coupled organic heterojunctions occurs on sub-20 fs timescales—approaching the half-period of high-frequency molecular vibrations—without requiring large energetic offsets or strong ground-state electronic coupling. Using synthetically defined donor–acceptor model heterojunctions, we demonstrate that spatial alignment of interfacially active vibrational modes enables vibronically driven, ballistic electron transfer. Ultrafast spectroscopy reveals electron transfer as fast as ~18 fs with accompanying coherent vibrational wavepackets on the acceptor, while quantum dynamical simulations complemented with experiments identify polymer-centered vibrational modes that mix Frenkel exciton (FE) and CT states, providing a mechanistic origin for this behavior.

## Results and discussion
### Model organic heterojunctions
We present a fundamental study of CT dynamics in a model heterojunction composed of donor and acceptor chromophores covalently tethered together. As shown in Fig. 1a, b, a symmetric and planar NFA, a perylene diimide (PDI) derivative[26], is covalently attached to a low-bandgap polymer (Ref-P), via non-π-conjugating alkyl chains in a "through-space" co-facial manner. Two different arrangements of the donor polymer and acceptor chromophores are explored, with the PDI unit at the face of either the electron-rich (BDT[27]) or electron-deficient (DPP[27]) moiety of the polymer backbone, which we will refer to as TS-P2 (acceptor-on-electron rich system) and TS-P3 (acceptor-on-electron deficient system) architectures, respectively (Fig. 1b, for synthetic protocols see "Methods" and ref. 28). These model systems allow us to isolate the dynamics of the CT process, which can be masked via exciton and charge-transport processes in OPV thin-films or by molecular diffusion in photocatalytic systems[29–32]. We emphasize that this study aims at elucidating the fundamental interfacial processes governing initial charge-separation dynamics, rather than optimizing device-level power-conversion efficiency. The model heterojunctions also allow us to probe the effect of the relative spatial orientation of different parts of the chromophores with respect to each other, something that would be impossible to study within a conventional OPV blend or photo-redox active heterojunction nanoparticle system[5,6] with disordered arrangements of chromophores.

Recently, it was demonstrated that in these model systems the microscopic orientation between donor and acceptor chromophores could significantly influence CT energies[28] and that the CT rates varied from slow (>1 ps) to fast (temporally unresolved), approximately below the instrument response of the measurements (150 fs). However, the fundamental molecular-level origin of the dramatic differences in CT rates between different configurations remains unknown, and the broader question of what ultimately governs the rate of CT in organic heterojunctions remains unanswered.

### Sub-vibrational period charge transfer
To study the CT dynamics of these model heterojunctions, we begin by employing pump-probe optical spectroscopy. As depicted in Fig. 1c, d The low band-gap Ref-Polymer has an absorption maximum at 750 nm, whereas the PDI electron acceptor has its peak absorption at 550 nm. The origin of the different intensities of the 0–0 and 0–1 peaks in polymer backbone chromophore absorption spectra has been discussed previously[28], and we believe that these differences predominantly arise from a greater degree of conformational disorder in TS-P3 relative to TS-P2. This separation in absorption features between the polymer backbone chromophore and PDI means that our sub-12 fs red pump pulse (Fig. 1e) excites only the donor polymer backbone, while the green pump pulse excites the PDI moiety predominantly due to its preferential absorption. The absorbance of the model heterojunctions and Ref-P measured with photothermal deflection spectroscopy (PDS) is presented in Fig. 1f.

We monitor the initially photo-excited polymer excitons with an infrared probe pulse after selectively pumping the polymer backbone chromophore with 700 nm narrowband optical pulses (pulse width 150 fs). As shown in Fig. 2a, the transient absorption (TA) spectra of the Ref-P polymer (turquoise green spectra) show a negative transmission signal with broad spectral coverage (1220–1580 nm), which can be assigned to the photoinduced absorption (PIA) of the polymer backbone neutral $S_1$ to $S_n$ state. The transient spectra of the TS-P2 (violet) and TS-P3 (orange) show an additional feature around 1250–1400 nm along with the neutral chromophore $S_1 \rightarrow S_n$ PIA of the backbone chromophore. This new blue-shifted spectral feature can be assigned to the hole absorption of the polymer (as confirmed by blending the Ref-P with the high (380 meV) PCBM/C60 LUMO energy offset acceptor, see Supplementary Note 3). The transient kinetic traces show attenuation of the neutral $S_1$ state population of the polymer backbone, indicating electron transfer to the PDI unit, as shown in Fig. 2b. While TS-P2 exhibits electron transfer on a sub-picosecond timescale, TS-P3 undergoes rapid quenching of its neutral excitonic state almost concurrently with photoexcitation—within the instrument response (~150 fs)—rendering the process temporally unresolved. In both TS-P3 and TS-P2, the hole wave function of the lowest energy excited state is predominantly localized on the DPP segment[28], resulting in a greater spatial separation between the electron (localized on PDI) and hole for the case of TS-P2. This geometry in TS-P2 leads to weaker electronic coupling and consequently a slower, configuration-dependent CT process compared to the TS-P3 architecture[28]. To investigate how fast the CT is in TS-P3, we excited the polymer backbone chromophore with a sub-12 fs 700 nm centered broadband red pulse (Fig. 1e, Supplementary Note 4), and the photoexcited species were probed with a temporally delayed broadband visible pulse, as depicted in Fig. 2d–f. As shown in Fig. 2d, for Ref-P, we observe ground state bleach (maxima at 760 nm), stimulated emission (SE) (820 nm), and a negative transmission signal around 900 nm, corresponding to the PIA of the $S_1$ state of the polymer backbone chromophore. These features exhibit minimal temporal dynamics in the very early time frame (20–40 fs) as expected for Ref-P.

In contrast, as shown in Fig. 2e, TS-P3 displays significant early time (20–40 fs) dynamics. The SE band and ground state bleach band of the polymer backbone chromophore are rapidly quenched within the first few temporal frames (<40 fs) due to the generation of an underlying negative differential transmission ($\Delta T/T$) signal. This negative signal is attributed to the absorption of the PDI radical anion, the photoproduct of the ultrafast electron transfer from the polymer backbone, which has a broad absorption spectrum across the probe wavelengths (illustrated by the blue shaded transmission spectra of PDI radical anion, for more details see Supplementary Note 5). Turning to TS-P2, as shown in Fig. 2f, the dynamics of the spectral features at the very early time frames (20–40 fs) are silent like in Ref-P. The

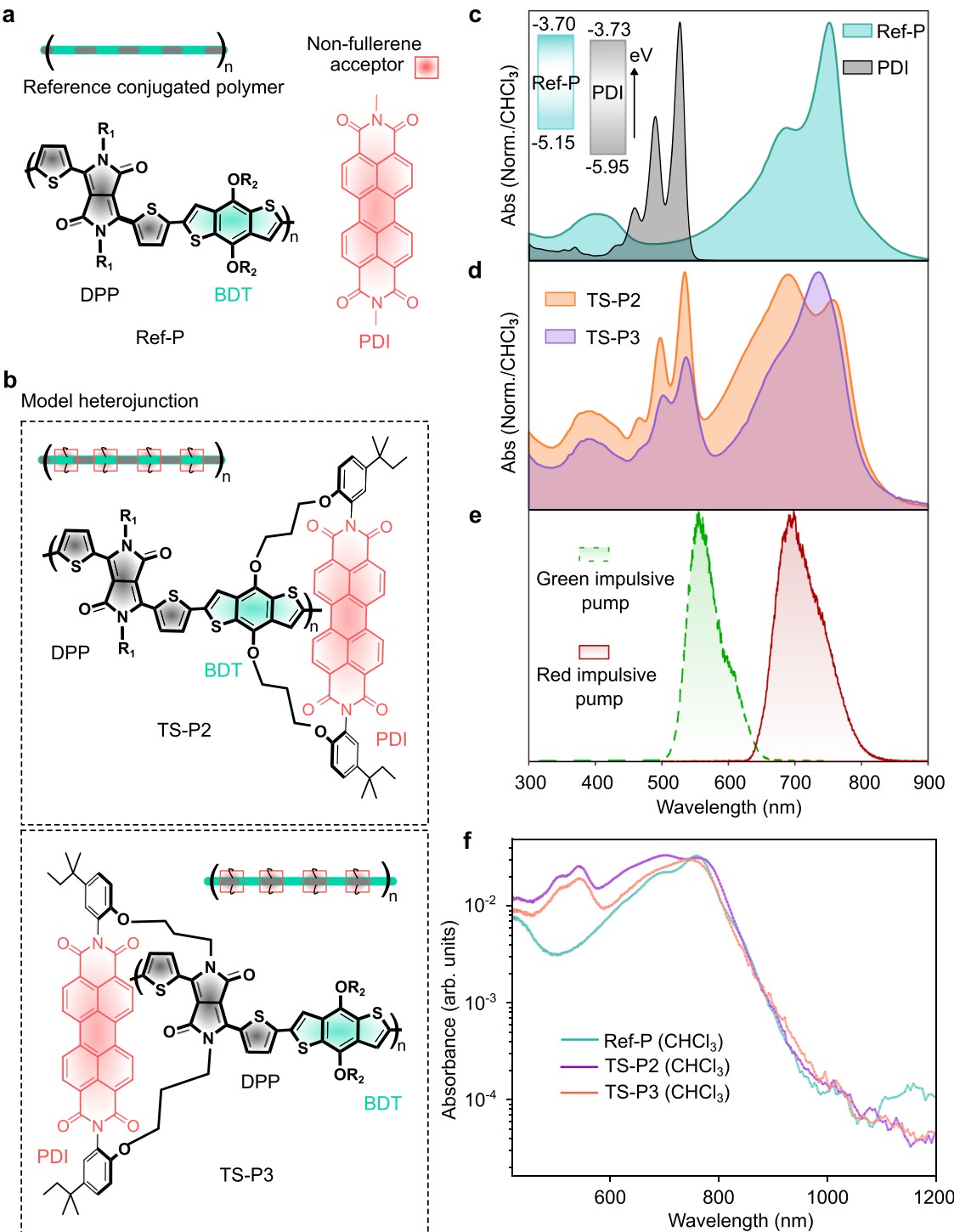

**Fig. 1 | Model organic heterojunction. a** Chemical structure of the donor polymer (Ref-P) and the symmetric and planar non-fullerene acceptor (NFA) unit, perylene diimide (PDI), used in the study. Ref-P is a BDT-DPP-based polymer. BDT stands for Benzodithiophenes, and DPP stands for diketopyrrolopyrrole. **b** Two model heterojunctions: Ref-P covalently linked with PDI with alkyl chains in a through-space "face-to-face" manner. Chemical structure of TS-P2, when PDI is closer to the BDT moiety of the DPP–BDT repeating unit, and TS-P3, when PDI is closer to the DPP unit. **c** Optical absorption spectra of the individual Ref-P and PDI in chloroform. The energy levels of the frontier molecular orbitals of Ref-P and PDI are illustrated, see Supplementary Note 1 for details. **d** Optical absorption spectra of the model heterojunctions, TS-P2 and TS-P3. **e** The green (dotted line) and red (solid line) impulsive sub-10-fs pulse used in the ultrafast transient absorption and broadband impulsive vibrational spectroscopic study. **f** Photothermal deflection spectra of the Ref-P, TS-P2, and TS-P3 measured in solution phase (CHCl₃) to mimic the liquid phase experiments.

photogeneration of the PDI radical anion happens in a much slower time scale (376 fs, Supplementary Fig. 42) in TS-P2 compared to TS-P3, Fig. 2c. The slower electron transfer rate is consistent with the previously described 150 fs IR-probed TA results where the TS-P3

dynamics remained unresolved. To quantify the ultrafast CT dynamics in TS-P3, kinetic analyses were performed after isolating the coherent artifact (Supplementary Fig. 41 and Note 17), which yielded a fast component of 18.1 ± 3.1 fs, corresponding to a conservative upper

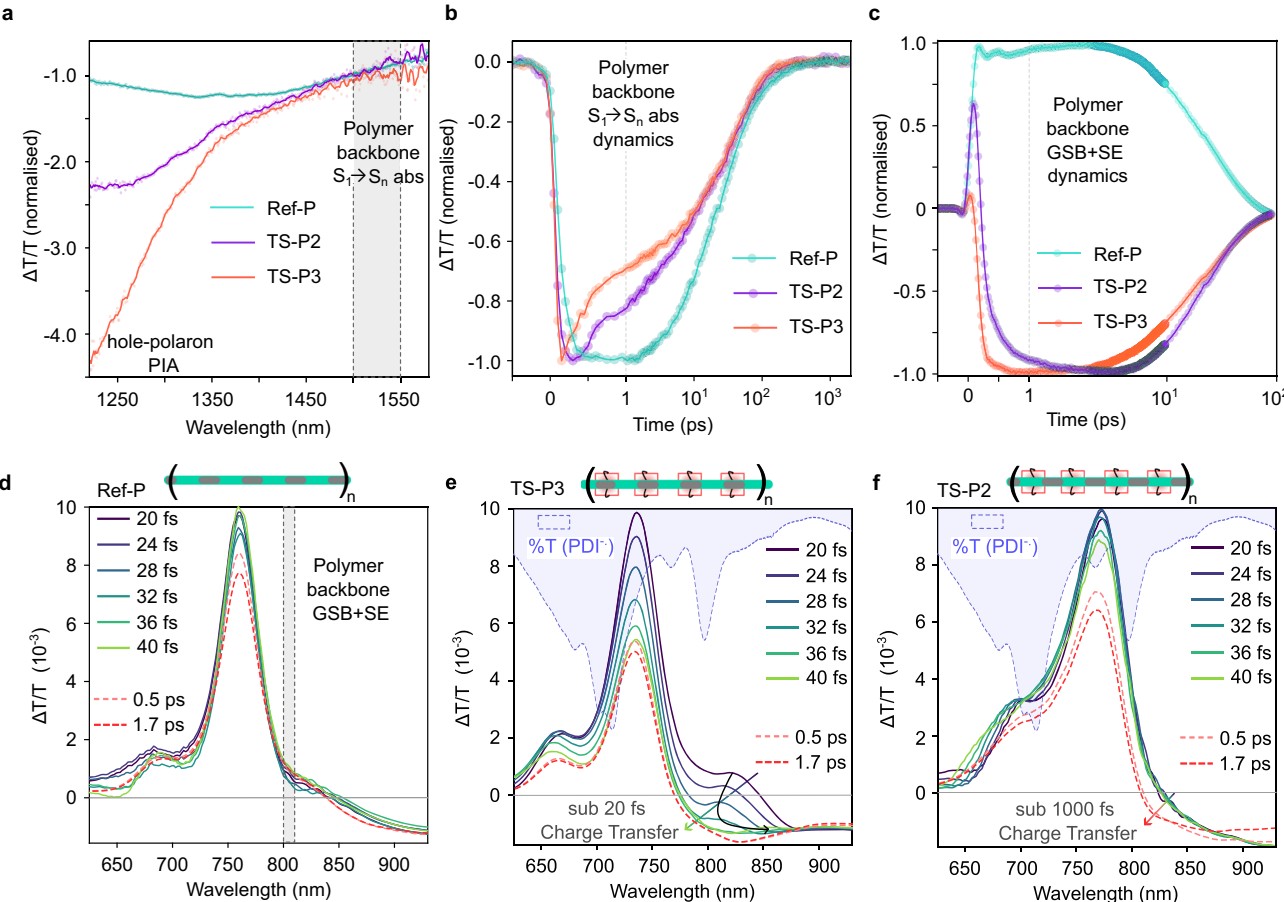

**Fig. 2 | Charge transfer dynamics in model heterojunctions. a** IR-probed differential transient absorption (TA) spectra of the Ref-Polymer, TS-P2, and TS-P3 polymer (in chloroform) averaged over 0.5–1.5 ps (normalized at 1500 nm) after photoexciting the polymer backbone chromophore with 700 nm, 150 fs pump pulses. **b** Population dynamics of the photoexcited neutral polymer backbone chromophore: TA kinetics extracted from the probe wavelength covering the photoinduced absorption ($S_1 \rightarrow S_n$) of the neutral polymer backbone singlet, ranging from 1500 to 1550 nm (gray area in **a**). **c** The picosecond temporal dynamics obtained for the 800–810 nm spectral range (gray shaded regime in (**d**) for Ref-Polymer, TS-P2, and TS-P3 polymer (CHCl₃ solution); photo-excited with 700 nm, 150 fs pulses. **d** Visible probe differential TA spectra of the Ref-Polymer (in chloroform) after photo-exciting the polymer backbone with 700 nm centered 10 fs "red impulsive pump." Visible probe differential TA spectra of **e** TS-P3 and **f** TS-P2 (in chloroform) after photo-exciting the polymer backbone with the 700 nm centered 10 fs "red impulsive pump." The blue shaded regime of (**e**, **f**) is the % Transmission spectra of the PDI radical anion (PDI⁻) in chloroform (see "Methods"). For wavelength and time-resolved data, see Supplementary Note 7.

bound of −21.2 fs for the electron-transfer process. Such sub-20 fs CT is markedly faster than in conventional OPV systems, where CT times are typically well above 15 fs, often by more than a few orders of magnitude[17–20]

Such rapid CT could be observed due to direct excitation of the ground state complex between the polymer backbone and the PDI, that is, of a CT transition, which may serve as a precursor for the ultrafast electron transfer. However, as demonstrated in Fig. 1f, PDS performed in chloroform solutions of the studied model heterojunctions does not reveal any broadening of the absorbance of TS-P3 with respect to Ref-P and TS-P2. A spectral broadening would correspond to the tail of broad, featureless near-bandgap CT transitions. This finding suggests that there is no directly optically excitable CT transition underlying the $\pi \rightarrow \pi^*$ transition of the polymer backbone chromophore. Furthermore, time dependent Density Functional Theory (TD-DFT) calculations performed on molecular models of TS-P3 (Supplementary Fig. 2) suggest sub band-gap ground state CT transitions to be two to three orders-of-magnitude weaker than the lowest polymer excitonic absorption in the Franck−Condon region, supporting the view that the red impulsive pump (Supplementary Note 2 and S13) prepares a pure $S_1$ state, which then undergoes a sub-20 fs electron transfer reaction.

We also note that the EL spectra of the device based on TS-P3 show that EL and PL spectra are closely matched, again suggesting that the $S_1$ and CT states lie close in energy[28]. In contrast to the rapid timescale of photoinduced electron transfer, excitation of the PDI unit preferentially leads to hole transfer to the polymer backbone, which occurs much more slowly (~200 fs), despite the energy offset for hole transfer being approximately 2.7 times greater than that for electron transfer (Supplementary Note 9).

**Charge-transfer-driven fast vibrational coherence generation**

From the results so far, electron transfer in TS-P3 occurs on a timescale similar to the period of high-frequency vibrations in this system. A question that arises is whether these vibrations play a role in this process. To gain a deeper understanding of how vibrational dynamics are coupled to the ultrafast CT reaction, we turned to impulsive vibrational spectroscopy (IVS)[33–36]. In resonant IVS (res-IVS), we used an ultrafast (sub-12 fs), spectrally broadband pulse centered at 700 nm (red impulsive pump, Fig. 1e) resonant to the polymer backbone $\pi \rightarrow \pi^*$ transition, which initiates the quantum superposition of vibrational states in the photoexcited state ($S_1$). This vibrational coherence evolves over time in accordance with the potential energy surface (PES) of the excited state. The resulting vibrational coherence records

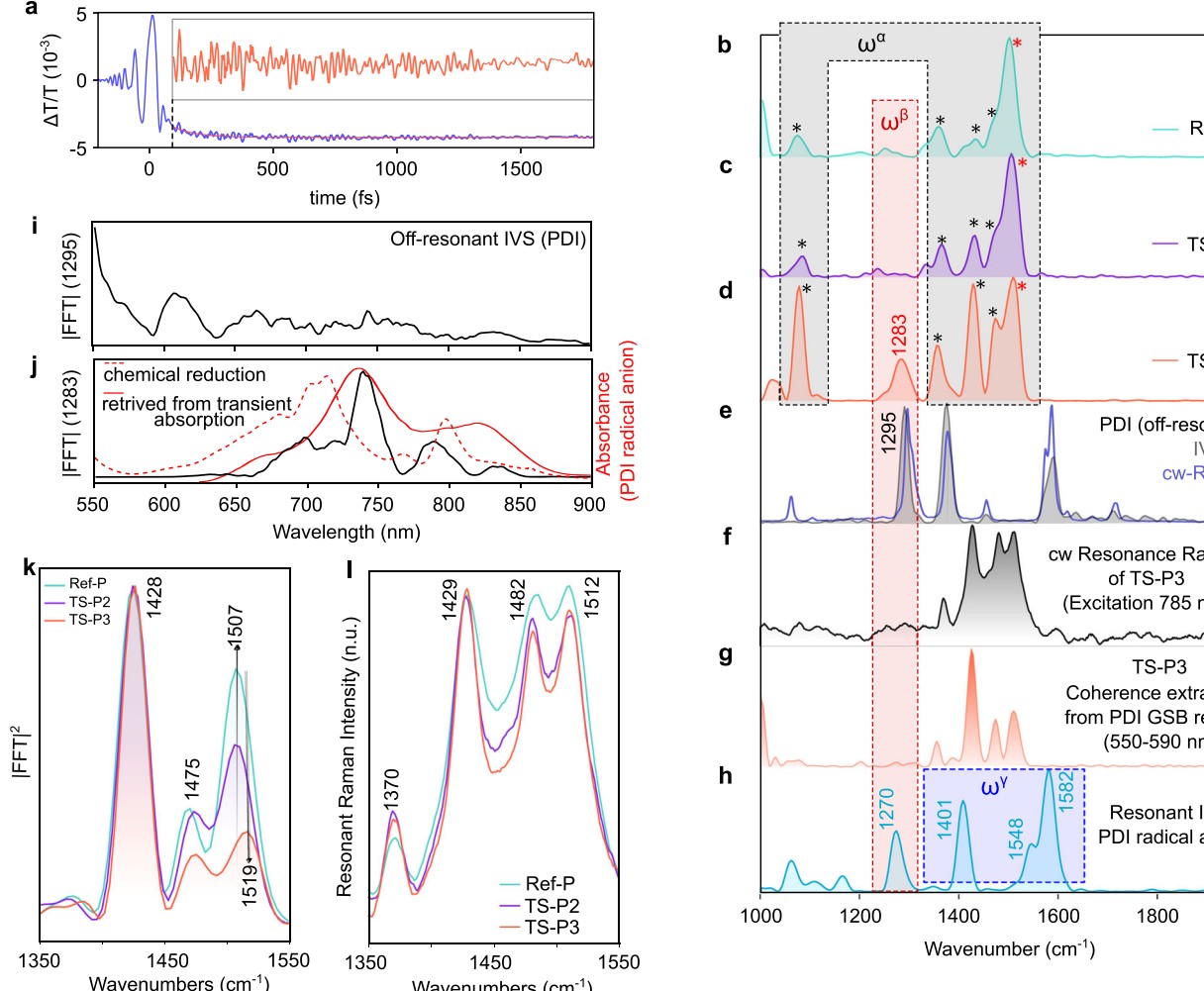

**Fig. 3 | Generation of the vibrational superposition state along the coordinate of the high-frequency quantum modes by electron transfer. a** Vibrational wavepacket motion overlaying the kinetic trace (in blue) and pure wavepacket dynamics after isolating the electronic contribution (in orange, inset) of the TS-P3 after photoexciting at the π → π* transition of the polymer backbone. The kinetic trace is extracted from the 790–830 nm probe region where the spectral feature converts from the stimulated emission of the polymer backbone to the PDI radical anion absorption on a sub-15 fs timescale. FFT power spectra of: **b** Ref-P (turquoise), **c** TS-P2 (purple), **d** TS-P3 (orange) integrated over the 790–810 nm probe regime in the high frequency regime (1000–2000 cm⁻¹). The "red impulsive pump" is used to create the coherent superposition states. **e** Gray spectra show the off-resonant IVS spectra of neutral PDI (saturated solution in chloroform), with vibrational coherence generated in the S₀ state by the red impulsive pump. The blue line shows the off-resonant CW Raman spectrum of neutral PDI (785 nm excitation). **f** Resonant CW Raman spectra of TS-P3 (excitation: 785 nm). **g** FT power spectrum of the

vibrational coherence obtained at the ground state bleach regime (550–590 nm) after photoexciting the polymer backbone in TS-P3 with the "red impulsive pump" (Supplementary Note 13, for details). **h** FT power spectrum of resonant IVS of the chemically reduced PDI radical anion (red impulsive pump), extracted from the ground-state bleach at 800–850 nm. **i** FT amplitude of the 1295 cm⁻¹ vibrational coherence vs probe wavelength under off-resonant PDI excitation with a red impulsive pump. **j** Black: FT amplitude of the 1283 cm⁻¹ vibrational coherence vs probe wavelength under resonant TS-P3 excitation with a red impulsive pump. Red dotted: PDI radical anion absorbance. Red solid: PDI anion absorption retrieved from TA (Supplementary Note 5). **k** CW-Resonant Raman spectra of the Ref-P, TS-P2, and TS-P3 at the high frequency regime (excitation wavelength, 785 nm). **l** Integrated Fourier transform IVS spectra wavelength range, 890–920 nm, which corresponds to the polymer backbone chromophore S₁ → Sₙ photo-induced absorption.

the system's impulsive response and appears as an oscillatory modulation superimposed on the transient population dynamics of the sample's excited-state transitions, as shown in Fig. 3a for TS-P3. Performing a Fourier transformation on these periodic modulations yields the excited-state Raman spectrum (Fig. 3b–d).

In Fig. 3b–d, we present the integrated Fourier-transformed vibrational spectra for the TA spectral region between 790 and 830 nm, which corresponds to the ground-state bleach (GSB) and the SE band of the polymer. This is shown for the Ref-P, TS-P2, and TS-P3, respectively. We note that for TS-P3, the rapid photoinduced CT means that this spectral region also contains the signature of the PDI radical anion absorption following CT. The Ref-P polymer's SE band has notable Frank–Condon coupling to bond-stretching/aromatic breathing $\{\omega_i^\alpha\}$

modes (where $\omega^\alpha$ (cm⁻¹) are 1060, 1368, 1433, 1474, 1507, asterisked modes in Fig. 3b) with the strongest intensity at 1507 cm⁻¹. The S₁ potential surface of the polymer backbone chromophore is displaced with respect to the S₀ potential surface along the coordinates of the $\{\omega_i^\alpha\}$ normal modes, hence photoexcitation gives rise to the generation of $\{\omega_i^\alpha\}$ wavepacket motions for Ref-P and also the model heterojunctions TS-P2 (Fig. 3c) and TS-P3 (Fig. 3d). Intriguingly, in the case of model heterojunction TS-P3, but not for TS-P2, a new vibrational coherence emerges with a frequency of 1283 cm⁻¹, which we refer to as $\omega^\beta$ (Fig. 3d). We can assign this newfound $\omega^\beta$ vibrational coherence to a high-frequency C–C stretching mode of the PDI acceptor moiety, since we observe a mode at similar frequency regime for an isolated PDI molecule with CW Raman spectroscopy (1295 cm⁻¹, blue spectra, Fig. 3e) and off-

resonant IVS (1291 cm$^{-1}$, gray spectra, Fig. 3e). In addition, the polymer backbone chromophore has no mode in this region 1275–1325 cm$^{-1}$ further confirmed by vibronic simulations with displaced harmonic oscillator model (Supplementary Fig. 50).

Below, we discuss evidence that this unique $\omega^\beta$ wavepacket is most consistently interpreted as arising following ultrafast electron transfer, rather than from direct photoexcitation of the acceptor.

First, we rule out the possibility that $\omega^\beta$ wavepacket is an off-resonant wavepacket launched on the ground-state surface of PDI, i.e., the possibility that the ultrashort optical pulse, off-resonant with the electronic transition of PDI, can also induce vibrational coherence on the ground-state surface of PDI. This phenomenon, which is known as impulsive stimulated Raman scattering[37] (ISRS), is also referred to as the generation of an off-resonant impulsive vibrational wavepacket[34]. To determine if the 1283 cm$^{-1}$ vibrational coherence is due to off-resonant excitation of wavepacket motion on the ground state manifold of the PDI units on TS-P3, we conducted an off-resonant IVS experiment. As shown in Fig. 3e (gray area), the FT spectra of the off-resonant wavepacket on the S$_0$ state of PDI (saturated solution of PDI in chloroform) induced by a red impulsive pump reveal a coherence at 1295 cm$^{-1}$, distinct from $\omega^\beta$. The wavelength-dependent amplitude of the off-resonant wavepacket at 1295 cm$^{-1}$ is depicted in Fig. 3i, showing the strongest activity in the <650 nm region. The presence of ground electronic state vibrational coherences near the tail of the optical absorption spectrum (<650 nm) is due to enhanced IVS signals as the probe pulse becomes near-resonant with an electronic absorption. This is a fingerprint of an off-resonant wavepacket, further confirming the assignment of the 1283 cm$^{-1}$ coherence (details in Supplementary Note 12). In contrast, the vibrational coherence at 1283 cm$^{-1}$ ($\omega^\beta$), obtained by resonant impulsive excitation of the polymer backbone in TS-P3, shows its strongest amplitude at probe wavelengths between 650 and 850 nm (black trace, Fig. 3j). This spectral envelope closely matches the PDI radical-anion feature retrieved from the TA data (red solid trace, Fig. 3j), and is further qualitatively complemented by the absorption profile of the chemically reduced PDI$^-$ species (red dotted trace, Fig. 3j).

The spectral localization of the 1283 cm$^{-1}$ vibrational coherence within the absorption of the PDI radical anion, rather than in the ground state bleach regime of PDI, suggests that the $\omega^\beta$ vibrational coherence signal is resonantly amplified by exciting the photogenerated radical anion. This indicates that the vibrational coherence is associated with the PDI radical anion on the excited-state CT state, instead of the ground electronic state of PDI. Additionally, the absence of an $\omega^\beta$ vibrational coherence for both TS-P2 (Fig. 3c) and chemical blends of Ref-P and PDI (Supplementary Fig. 23), despite both systems having identical PDI content per polymer backbone chromophore, provides further evidence that the $\omega^\beta$ (1283 cm$^{-1}$) wavepacket does not stem from an off-resonant vibrational coherence.

We next discuss why the $\omega^\beta$ wavepacket is not an outcome of the direct CT excitation across the donor–acceptor heterojunction. Utilizing continuous wave resonance Raman with 785 nm excitation on TS-P3, we do not observe the $\omega^\beta$ (1283 cm$^{-1}$) mode (Fig. 3f), even though the PDI modes—including the analogous 1291 cm$^{-1}$ band—exhibit strong Raman cross-sections when excited resonantly (Supplementary Note 19). This observation eliminates the possibility of a ground-state CT complex between PDI and the polymer backbone, which could be directly photoexcited impulsively. Furthermore, during the impulsive excitation of the polymer backbone in TS-P3, we are unable to detect any vibrational coherence at $\omega^\beta$ within the PDI Ground State Bleach (GSB) regime (550–580 nm), as shown in Fig. 3g. The mode is only observed in the spectral region (650–850 nm) where the PDI anion shows a PIA.

Finally, we discuss why $\omega^\beta$ wavepacket is not a re-excitation of the photogenerated PDI anion. Considering the remarkably rapid electron transfer (sub-15 fs) observed from the polymer backbone chromophore to PDI in TS-P3, it is crucial to acknowledge that the 1283 cm$^{-1}$ ($\omega^\beta$) wavepacket mode could be impulsively generated by direct photoexcitation of the promptly photogenerated PDI radical anion during the course of the photoexcitation (sub-12 fs) of the TS-P3 polymer backbone. To investigate this possibility, we synthesized the chemically reduced ground-state open-shell PDI radical anion to mimic the photoproduct (using tetraamine as the base, see "Methods") and conducted resonant IVS by photoexciting with an identical 700 nm centered broadband red impulsive pump. As depicted in Fig. 3h, we found that the photoexcited PDI radical anion (PDI$^{-\cdot}$*) has a strong coupling of the $\{\omega_i^\gamma\}$ modes (where $\omega^\gamma$ are 1401, 1544, and 1584 cm$^{-1}$) to the PIA and GSB transitions, along with relatively weaker coupling at 1266 cm$^{-1}$ ($\omega^\beta$) mode (for details, Supplementary Note 11). Notably, the 1266 cm$^{-1}$ mode shows the maximum displacement in comparison to the corresponding mode in neutral PDI ($\Delta\omega = 29$ cm$^{-1}$), in comparison to $\Delta\omega_i^\gamma = 21$ cm$^{-1}$ (for 1401 cm$^{-1}$), 7 cm$^{-1}$ (1544 cm$^{-1}$), 8 cm$^{-1}$ (1584 cm$^{-1}$). Therefore, during the impulsive excitation of the polymer backbone in TS-P3 (Fig. 3d), the absence of the $\{\omega_i^\gamma\}$ modes, which are strongly coupled to the resonant impulsive excitation of PDI$^-$ → PDI$^{-\cdot}$* but lack pronounced frequency shifts from neutral to anionic potential surfaces like the $\omega^\beta$ wavepacket, suggests that $\omega^\beta$ is not an outcome of the in-situ excitation of photogenerated PDI charges. See Supplementary Fig. 44 for a decision-tree summary of the evidence supporting the assignment of the 1283 cm$^{-1}$ coherence.

The comprehensive set of control measurements and observations presented above suggests that the vibrational coherence at the $\omega^\beta$ (1283 cm$^{-1}$) mode, observed exclusively during the electron transfer in the TS-P3 heterojunction, does not emerge following impulsive excitation of PDI with the ultrashort pump pulse. Therefore, the $\omega^\beta$ coherence observed here is most consistent with an origin associated with the ultrafast electron transfer process itself.

In our model heterojunctions, the PDI acceptor molecule vibrates along the coordinates of the normal modes with defined natural frequencies, before any electron transfer occurs. It is important to note that most of these normal modes have a vibrational period slower than 15 fs. However, upon photoexcitation of the model heterojunction TS-P3, an ultrafast electron transfer event occurs on a timescale of less than 20 fs (see Supplementary Fig. 41). This rapid electron transfer creates an impulsive shift[38,39] of the nuclear coordinates where the PES of the PDI radical anion is displaced relative to that of the neutral PDI. Consequently, in these PDI molecules, vibrational coherences are impulsively generated, transitioning to the newly displaced vibrational mode. This collective response manifests as coherent vibrational motion along the coordinate of $\omega^\beta$. The short-time FT analysis of the vibrational coherence (Supplementary Fig. 47) shows a clear temporal lag in the appearance of the $\omega^\beta$ mode, further suggesting that it arises from CT dynamics rather than direct impulsive excitation.

To further explore these experimental observations, we constructed a linear vibronic coupling (LVC) model for a representative TS-P3 monomeric unit, including the most relevant vibrational modes that couple to the localized (diabatic) FE state on the DPP–BDT unit as well as to two CT states (in which the electron is transferred from DPP–BDT to PDI). Details are provided in "Methods" and Supplementary Note 21. Using this model, we performed full quantum dynamics simulations with the multi-layer multi-configurational time-dependent Hartree method[40]. The simulations reveal that the PDI vibrational mode at 1299 cm$^{-1}$—coincident with the experimentally observed $\omega^\beta$ feature in Fig. 3d—begins to evolve only after the CT states become populated through ultrafast electron transfer (see Supplementary Fig. 51). In contrast, the DPP–BDT polymer modes start oscillating immediately at time zero and act as driving modes for the CT as we discuss below. We note in passing that several groups have attempted similar excited state dynamics simulations, using not only full quantum dynamics approaches but also mixed quantum–classical molecular dynamics[41–43].

We could experimentally capture this vibrational coherence, as the corresponding time period of the $\omega^\beta$ mode ($\tau$: 24 fs) is slower than the convoluted timescale of photoexcitation and electron transfer $\left( < \sqrt{(12^2 + 15^2)} = 19.2\,\text{fs} \right)$. The electron transfer observed here generates vibrational coherence similarly to what has been previously observed in solvent to dye[38] ET ($\tau$: 106 fs), intramolecular ET in dye[44] (250 fs), and ET to inorganic colloids[45] (230 fs), lending further support to our interpretation. Generation of an ET-driven coherence for a vibrational mode with a period of a 26 fs period thus provides an internal marker that is consistent with configuration-dependent ultrafast electron transfer occurring on a sub-vibrational oscillation time scale.

## Vibrational modes that drive charge transfer

As shown in Fig. 3b–d, apart from the appearance of the mode at 1283 cm$^{-1}$ discussed above, there are also major differences in the relative intensities of the high-frequency modes between Ref-P, TS-P2, and TS-P3. In TS-P2, where the CT is slow compared to TS-P3, we observe an identical intensity pattern for the vibrational modes associated with the polymer in comparison to the Ref-P Frank–Condon active {$\omega_i^\alpha$} wavepacket, asterisked modes in Fig. 3b, c. In comparison, the relative intensities of the modes change considerably in the case of TS-P3.

The data presented in Fig. 3b–d are taken from the spectral region associated with the polymers GSB (790–830 nm). To exclude the possibility that vibrational coherence being observed in this region could be mixed with the ground state vibrational coherence generated by the ISRS[34,37,46], we also analyze the PIA region (890–920 nm) associated with the $S_1 \rightarrow S_n$ transition in the polymer. As clearly seen in Fig. 3k, there is a significant loss in FFT power for the mode at 1507 cm$^{-1}$ for TS-P3 in comparison to TS-P2 and Ref-P. In contrast, the CW-Resonant Raman spectra of the three systems show nearly identical features, Fig. 3l. This means that the loss in strength of the 1507 cm$^{-1}$ for TS-P3 stems from vibrational decoherence from the processes/reactions happening in the excited state landscape, similar to as observed before for small molecules in solution[38,47]. This is further supported by the faster decoherence of the 1507 cm$^{-1}$ mode in TS-P3 relative to Ref-P, whereas the other Franck–Condon-active modes exhibit comparable dephasing times in both systems (Supplementary Fig. 46) in inverse FT analysis. Based on extensive excited-states TD-DFT calculations (see below), we hypothesize that this decoherence stems from the ultrafast (<15 fs) CT process observed in TS-P3, and that the 1507 cm$^{-1}$ mode is a driving mode, i.e., a vibrational mode that drives an ultrafast chemical reaction. This is in contrast to spectator modes (e.g., 1429 cm$^{-1}$), which are not affected by the ultrafast chemical reaction and hence do not undergo additional decoherence apart from natural decoherence. We further note that the time period of this 1507 cm$^{-1}$ mode, 22 fs, is consistent with the timescale of the ultrafast photoinduced electron transfer reaction we observe.

To assess the nature and the role of relevant vibrational modes in the excited-state dynamics of TS-P3, we performed Density Functional Theory (DFT) calculations on a representative molecular dimer, offering a reasonable balance between accuracy and computational cost (see Supplementary Note 15). The calculated Raman spectrum reported in Supplementary Fig. 29 shows excellent agreement with experiment. Namely, in the relevant spectral range between 1400 and 1550 cm$^{-1}$, the calculations yield three intense bands peaking at 1441, 1495, and 1529 cm$^{-1}$ matching closely the frequencies of the observed three main bands in Fig. 3 (respectively at 1429, 1482, and 1512 cm$^{-1}$). These three vibrational modes involve mostly the polymer backbone with different weights on the DPP vs BDT units, see Supplementary Note 15. It is noteworthy that the 1529 cm$^{-1}$ mode is almost fully confined on the electron-poor DPP moieties facing the PDI acceptors.

We then performed excited-state TD-DFT calculations on structures that were generated by distorting the TS-P3 dimer along the relevant high-frequency modes. The results are reported in Fig. 4 (see also Supplementary Note 15), where we show the adiabatic PESs constructed for the lowest 8 excited states by considering finite positive and negative displacements along the 1440, 1441, 1495, 1527, and 1529 cm$^{-1}$ vibrational modes. In the Franck–Condon region, that is at the ground-state equilibrium geometry (zero displacement on Fig. 4), the low-energy spectrum comprises a few CT excitations (CT-1 to CT-4) located below the optically allowed FE state, and is almost dark (see also Supplementary Fig. 36, where natural transition orbitals are shown). More specifically, if all four states involve the same lending PDI-centered orbitals (where the electron resides), the most strongly bound CT-1 and CT-2 states have a dominant hole contribution associated with the DPP units located in close proximity with the PDI cores, while CT-3 and CT-4 are longer-range CT states with the hole wave function primarily located on the nearby BDT units (hence their higher energy). Before proceeding, it is important to stress that the PES reported on Fig. 4 corresponds to adiabatic potentials; hence, the nature of the states might evolve when displacing the geometry along the normal-mode coordinates. The evolving character of the FE state informs how strongly it interacts with the nearby CT excitations and, as a proxy for that interaction, we focus on the FE oscillator strength, represented in Fig. 4a, b by the size of the full circles. Figure 4a, b shows completely different behavior when activating either the 1441 or the 1529 cm$^{-1}$ mode. In the former case (Fig. 4a), we observe a lowering in the FE energy upon relaxing the geometric structure along the corresponding normal-mode coordinate, but all the CT states are equally stabilized.

This results in the FE oscillator strength barely changing in the range of displacements explored. Hence, we conclude that the 1441 cm$^{-1}$ mode is a spectator mode. In contrast, the FE energy plunges when moving along the 1529 cm$^{-1}$ mode, leading to (avoided) crossings with the CT excitations and a massive reorganization of the oscillator strength, see Fig. 4b. In a way, this is expected since this vibrational mode has a large amplitude over the regions of the backbone in close contact with the PDI acceptors. Thus, the theoretical calculations support the view that the 1529 cm$^{-1}$ mode drives the electron transfer from the polymer backbone to the nearby PDI units, by causing a mixing of the singlet FE and CT PES surfaces. This is illustrated in Fig. 4c. Hence, it is this driving mode that undergoes a large reduction of intensity in the IVS spectra. This mechanism is conceptually analogous to the vibronic coherence-driven coupling between locally excited and CT states reported in symmetric molecular dimers showing SBCT[24,25], but here it operates within an asymmetric, weakly coupled heterojunction (i.e., ~25 meV in TS-P3 as shown in Supplementary Table S4).

Because the PDI is attached via a non-conjugated linker, CT in TS-P3 occurs through space. This is supported by the absence of CT in a Ref-P + PDI mixture (Supplementary Note 16) and by electronic coupling calculations performed on TS-P3 with and without the linkers (Supplementary Note 21), which produce nearly identical FE–CT couplings (Supplementary Table S5), confirming that linkers do not mediate the interaction. Thus, the ultrafast <15 fs CT arises uniquely from the tethered through-space geometry and its coupling to the DPP-localized vibrational mode.

The regio-specific contrast between TS-P3 and TS-P2 underscores the generality of this vibronic-driving mechanism. When the high-frequency DPP-localized vibration (1512–1529 cm$^{-1}$) coincides with the donor–acceptor interface, as in TS-P3, it directly modulates the CT gap, enabling sub-20 fs electron transfer. In TS-P2, where this mode is spatially displaced from the PDI acceptor, the coupling is weaker, and CT slows markedly ($\approx$376 fs). This demonstrates a general design rule: co-localizing interfacially active vibrational modes with the electronic coupling region enables ultrafast CT even in weakly coupled systems.

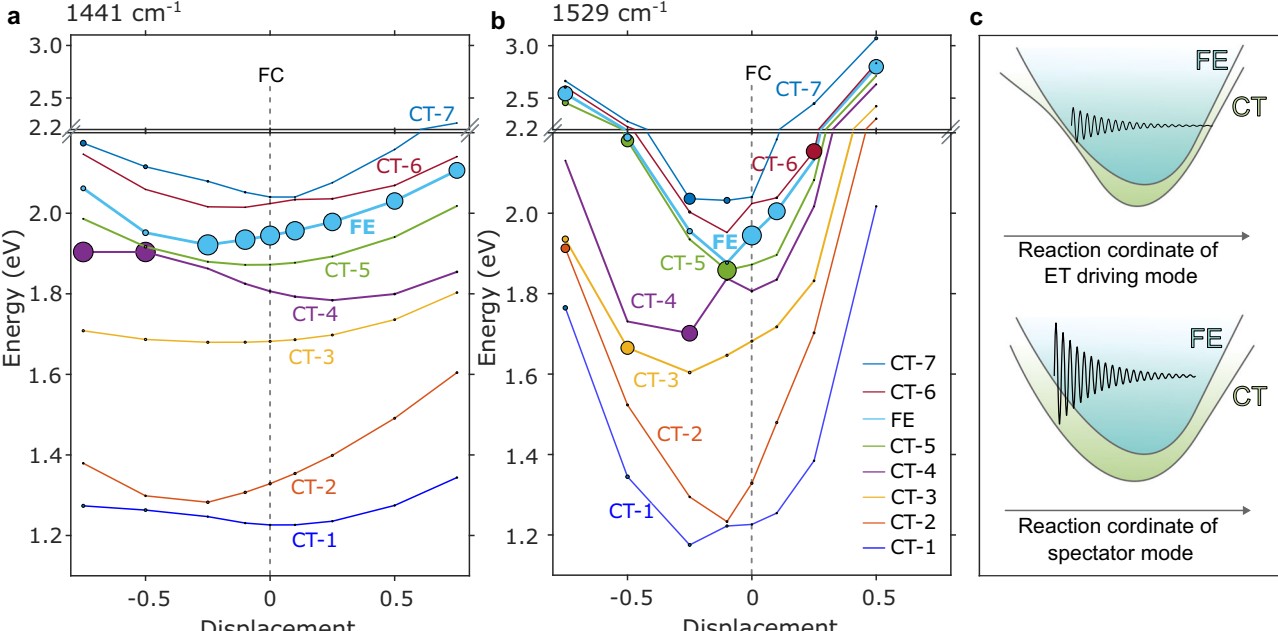

**Fig. 4 | Spectator versus driving vibrational modes from excited-state calculations.** Excited-state energy landscape upon finite displacements along the relevant high-frequency normal modes **a** 1441 cm$^{-1}$ and **b** 1529 cm$^{-1}$. The dotted black line indicates the Frank–Condon (FC) region, while the size of the full circles over the different PESs quantifies the oscillator strength of the bright Frenkel exciton state. **c** For an ET driving mode (top), motion along the vibrational coordinate strongly mixes FE and CT potential energy surfaces, causing ultrafast electron transfer and a rapid loss of vibrational coherence (decayed wiggles). For a spectator mode (bottom), FE and CT surfaces remain weakly coupled, so electron transfer does not perturb the wavepacket, resulting in persistent coherence.

Taken together, we have demonstrated that in a series of model donor–acceptor heterojunctions, the rate of CT can be precisely tuned over orders of magnitude through synthetic control of the spatial overlap between donor and acceptor moieties in polymers incorporating NFAs, achieving CT times as short as 15 fs. Such a fast CT rate in organic molecules occurs even though there is no strong coupling between donor and acceptor in the ground state, as evidenced by the unperturbed ground state absorption spectra of the constituent units. Nor is there a large energy offset between frontier orbitals to provide a large driving energy for the reaction. Instead, this process is activated by high-frequency vibrational driving modes that mix singlet (FE) and CT states following photoexcitation, allowing for CT to be achieved on timescales much faster than would be expected when considering purely static electronic coupling interactions. This implies that the half-period of such high-frequency modes sets the ultimate speed limit for the CT time in organic systems and that it can be accessed without the need for strong electronic coupling in the ground state.

Importantly, the ultrafast CT observed here is ballistic rather than diffusive: vibronic coupling launches the electron coherently along the reaction coordinate, driving transfer vectorially rather than relying on slow stochastic fluctuations to reach the CT geometry. These results have important implications for the design of future OPV, photocatalytic, and optoelectronic materials, where our results suggest that previously established design rules about the tradeoff between energy loss and CT rates may not represent prevailing limitations to improve further materials design strategies.

## Methods
### Material
The synthesis of the materials, Ref-P2, TS-P2, and TS-P3 has been described in detail in our previous report[28]. For solution phase optical spectroscopy measurement, an optical density of 0.25 is maintained at 700 nm for polymer solutions.

Chemical reduction of the PDI to produce the radical anion was carried out by systematically adding tetrakis(diethylamino)ethylene to the PDI solution (in CHCl$_3$) and followed by purging the N$_2$ gas inside solution to remove any residual O$_2$ similarly as reported earlier[38]. Formation of the open-shell PDI radical anion was monitored by UV-VIS absorption spectroscopy.

### Film and device fabrication
For the TA on the films, a stock solution of pristine and blend (1:1 wt% with PCBM/C60) Ref-P2 in [chlorobenzene (10 mg mL$^{-1}$)] was spin-coated (at a speed of 200 rpm) on 25 × 25 mm quartz substrate which was precleaned by subsequently sonicating in soaped DI water, DI water, Acetone and IPA and plasma-cleaned by O2 plasma. The spin-coated films were further encapsulated by a 170 μm glass slide with epoxy in an inert atmosphere.

### Device fabrication for ER-EIS measurements
Fabrication of electrodes started with 10 min of sonication of borosilicate glass wafers (900 μm-thick, double-side polished Microchemicals) submerged in acetone, followed by IPA, then baked at 150 °C to remove residual moisture. Gold contacts were patterned using metal lift off, which consisted of coating with AZ nLoF 2035 negative photoresist (Microchemicals) (spin coating at 500 rpm for 5 s, acceleration of 1000 rpm s$^{-1}$, followed by 3000 rpm for 45 s, acceleration of 8000 rpm s$^{-1}$, then soft baked at 110 °C for 60 s) followed by UV exposure (60 mJ cm$^{-2}$), a postexposure bake (110 °C for 180 s) and development in AZ 826 MIF (Microchemicals) for 30 s. Then, the photopatterned wafer was coated with 5 nm of titanium, then 100 nm of gold (E-beam Evaporator, Kurt J. Lesker Company), and lift off was performed by submerging the wafer in acetone for 1 h, followed by rinsing with acetone, then IPA. The patterned metal-coated wafers are then coated with a parylene bilayer by first treating the wafer with oxygen plasma for 60 s, followed by submerging the wafer in a dilute silane solution (3% v/v A174 silane dissolved in 0.1% v/v acetic acid in deionized water) for 45 s to improve parylene adhesion to the wafer. The silane treated wafer is rinsed with ethanol and heated for 1 h at 75 °C then coated with a 2 μm layer of parylene (PDS 2010 Labcoter 2,

Specialty Coating Systems) followed by coating with a soap surfactant layer (2% v/v Micro 90 soap in deionized water spin coated at 1000 rpm for 30 s and dried in air for 20 min) followed by a second deposition of a 2 μm layer of parylene. The trenches for depositing organic electrodes were defined in the parylene bilayer with photolithography by coating with AZ10XT positive resist (Microchemicals) (spin coating at 3000 rpm for 45 s, acceleration of 8000 rpm s⁻¹, soft baking at 115 °C for 120 s) followed by UV exposure (540 mJ cm⁻²) and developing in AZ 7 26 MIF developer (Microchemicals) for 10 min. Then, the parylene was etched using reactive ion etching (recipe) and the wafers were diced with a diamond scribe and tile cutter tool. Organic electrodes were coated. The organic electrodes were defined by peeling off the top parylene layer using Kapton tape, leaving the organic only in the patterned trench. A silicone well was defined using an adhesive-backed silicone (McMaster-Carr) to confine the electrolyte.

### ER-EIS measurements

Energy-resolved electrochemical impedance spectroscopy (ER-EIS) measurements were carried out following procedures as described in ref. 48. Measurements were carried out using 100 mM TBA:PF6 in anhydrous acetonitrile as the electrolyte, a silver-silver chloride reference electrode, and a platinum wire counter electrode (BASi Research Products). The active area of electrodes ranged from 9.1 to 0.05 mm², and amplitudes of the working electrode current were normalized by the electrode area. We used a sinusoidal working electrode perturbation with a frequency of 1 Hz and an rms amplitude of 100 mV. The mean potential of the working electrode was swept from 0 to −1.5 V and 0 to 1.5 V with 0.01 V increments. The frequency was chosen by carrying out a preliminary EIS scan and selecting a frequency where the rate-limiting step was charge exchange at the electrolyte/organic electrode interface. Forward and reverse voltage sweeps were carried out on different samples to avoid sample variations due to ion intercalation from affecting the measured band edges. The energy scale was calibrated using the ferrocene oxidation/reduction potential (E0 = 5.08 eV vs vacuum) measured by cyclic voltammetry (−1.0 to 1.0 V vs Ag/AgCl, sweep rate of 50 mV s⁻¹) carried out after ER-EIS measurements. The density of states, g($E\_F$), was extracted from the out-of-phase admittance, $Y''$(E), using the following Eq. (1):

$$Y'' = \omega C_{SC}(E_F) = e^2 L S g(E_F) \qquad (1)$$

Where $\omega$ is the angular frequency, $e$ is the elementary charge, $L$ is the sample thickness, and $S$ is the electrode area. For our analysis, we plotted the relative g($E\_F$) to find the HOMO and LUMO levels by fitting to a Gaussian g($E\_F$) model. Band edges were determined as the intersection between the tangent of the Gaussian and the background signal.

### Photothermal deflection spectroscopy

PDS on the solution samples was performed following our previously established protocols[49], using a tuneable light source consisting of a 250 W quartz tungsten halogen lamp coupled with a grating monochromator (Andor Kymera 328i) to produce the pump beam, further modulated with a mechanical chopper at 8 Hz. Polymer solutions in chloroform were poured into a 1-mm-wide quartz cuvette and placed in the focal spot of the pump beam. A 670 nm continuous-wave diode laser beam was passed through the solution volume, crossing the pump focal spot. The solvent acted as the thermooptic liquid, transforming the heat generation in the dissolved polymer to local fluctuations of the refractive index of the solvent. The concentration was chosen to maximize the generated signal while minimizing the absorption and scattering of the probe laser beam, at which a typical optical density at 670 nm ranged between 0.4 and 0.5 for all samples. The probe beam deflection was converted to a voltage signal by a quadrant photodiode

and demodulated with a lock-in amplifier (Stanford Research Systems SR830). PDS measurements were performed using the configuration described above, which provides all parameters required for replication. Similar implementations have been reported previously[49]. We chose the solution-phase method to ensure consistency with the solvent environment used in our TA and vibrational experiments, and to avoid additional aggregation or morphology effects that can complicate interpretation in thin-film measurements.

### Resonant Raman scattering

Resonance Raman measurements were performed on microcrystalline samples in a Renishaw inVia microscope with a 100× objective and <100 μW incident power. The laser excitation wavelength used was 785 nm. Photoluminescence backgrounds were removed by a restricted region polynomial fitting, and calibrated for the transmission efficiency of the measurement system.

### IR-probed (<200 fs) transient absorption spectroscopy

The broadband IR-probe time-resolved spectroscopy (1200–1650 nm) was conducted on a setup powered by a Ti:sapphire amplifier, which was obtained from Spectra Physics Solstice Ace. This amplifier operated at a 1 kHz rep-rate, producing 150-fs pulses centered at 800 nm with a 7 W output. For the short-time (100 fs–1.8 ns) time-resolved measurements, a TOPAS optical parametric amplifier was utilized to generate tunable 150-fs pump pulses. These pump pulses were also produced by a broadband visible (525–775 nm) non-collinear optical parametric amplifier (NOPA), seeded by a chirped white light source. The probe pulses were obtained from an infrared (1250–1650 nm) NOPA, and amplification was achieved at a PPSLT crystal. To detect the probe pulses, an InGaAs dual-line array detector (Hamamatsu G11608-512DA) was employed, controlled and read out by a custom-built board from Stresing Entwicklungsbüro. To minimize noise and correct for any shot-to-shot fluctuations in the probe, the probe beam was split into two identical beams using a 50/50 beam splitter. One of these beams served as a reference and also passed through the sample, but didn't interact with the pump. This reference beam allowed precise corrections to be made, enabling the measurement of very small signals with $\Delta T/T = 1 \times 10^{-5}$.

### Impulsive vibrational spectroscopy

The IVS was carried out using a home-built system seeded by a commercially available Yb:KGW amplifier laser (PHAROS, Light Conversion) operating at 1030 nm with a repetition rate of 38 kHz and an output power of 15 W following a previously published method[33]. To generate the probe pulse, we employed a chirped white light continuum (WLC) spanning from 530 to 950 nm. This WLC was produced by focusing a part of the fundamental beam onto a 3 mm YAG crystal and then collimating it. The impulsive pump pulses, as previously reported, were generated via non-collinear optical parametric amplification (NOPA). For the generation of the second (515 nm) and third harmonic (343 nm) pulses required to pump the NOPAs, an automatic harmonic generator (HIRO, Light Conversion) was used. In the experiments reported, the impulsive pump to excited PDI-dominated transitions was generated using a NOPA seeded by the 1030-WLC and amplified by the third harmonic (343 nm). On the other hand, for polymer backbone chromophore excitation, the pulse was generated using a NOPA seeded by the 1030-WLC and amplified by the second harmonic (515 nm). To compress the pump pulses, a pair of chirped mirrors in combination with wedge prisms (Layertec) were employed. We characterized the spatio-temporal profile of the pulses using second-harmonic generation frequency-resolved optical gating (SHG-FROG) (see Supplementary Fig. 6). To generate differential transmission spectra, a chopper wheel in the pump beam path modulated the pump beam at 9 kHz. The pump-probe delay was precisely controlled using a computer-controlled piezoelectric translation stage

(PhysikInstrumente) with a step size of 4 fs. Both pump and probe polarizations were set to be parallel. For our measurements with solution samples, we utilized a flow cell cuvette with an ultrathin wall aperture (Starna, Far UV Quartz) and a path length of 0.2 mm. To compensate for the dispersion effect produced by the cuvette wall, pulse compression was performed by placing a quartz coverslip (170 microns) in the beam path of FROG.

### Time-domain vibrational data analysis

After applying chirp-correction and background subtraction, we further refined the kinetic traces by truncating data points with time delays <100 fs to eliminate contaminations from coherent artifacts. Next, we isolated the residual oscillations by globally fitting the electronic dynamics using a sum of two exponential decaying functions with an offset across the entire spectral range. To transform the oscillatory time domain signals into the frequency domain, we employed a series of signal processing techniques, including apodization (Kaiser-Bessel window with $\beta = 1$), zero-padding, and Fast Fourier transformation. To ensure accurate results, we multiplied the |FFT| amplitude by a frequency-dependent scaling function to remove time-resolution artifacts before generating the intensity spectra.

### Computational modeling

The ground state structure of representative molecular TS-P3 dimer and monomer model systems was optimized at the DFT level with the ωB97X-D exchange-correlation functional and the 6-31G(d,p) basis set. Alkyl chains were removed to reduce the computational cost. The optimized structure was then subjected to a frequency analysis, along with a Raman spectrum calculation with the same level of theory, where frequencies were scaled by a factor of 0.949 to account for anharmonicity. The most intense Raman modes in the 1400–1550 cm$^{-1}$ region were analyzed by quantifying the relative weight of the displacement over the different molecular units of the TS-P3 dimer: BDT, DPP, and PDI (see Supplementary Table S2).

Time-dependent (TD) DFT calculations were then performed for the optimized TS-P3 dimer structure (i.e., at the Frank–Condon region) and for all the geometries obtained by displacing the structure (both in a positive and negative direction) along the relevant high-frequency normal modes (see Fig. 4 and Supplementary Fig. 36). The displacements, expressed in Å, were performed by using the *GaussView 6* graphical interface. All the TD-DFT calculations were run by using the *screened* range-separated hybrid (SRSH) approach[50], where the LC-ωhPBE functional was used, along with the 6-311G(d,p) basis set. In this scheme, where the interelectron Coulomb operator is partitioned between a long- and a short-range domain, the range-separation parameter ω was optimally tuned (OT) in gas-phase (at $\omega = 0.103$ Bohr$^{-1}$) and the dielectric constant of toluene ($\varepsilon = 2.37$) was set *a-posteriori* via the adjustable parameter β, while the parameter α (that is the fraction of the Hartree-Fock exchange amount in the short-range domain) was kept fixed at 0.2, so to satisfy the relationship for which $1/\varepsilon = \alpha + \beta$. All the (TD)-DFT calculations were carried out by using the Gaussian16 suite of packages[51].

A LVC Hamiltonian was constructed to model the ultrafast excited-state dynamics of a TS-P3 monomer model system. The Hamiltonian includes diabatic FE states localized on the DPP–BDT backbone and CT states involving electron transfer from DPP–BDT to PDI, together with selected vibrational modes and their electron–vibrational couplings. Diabatic energies and FE–CT electronic couplings were obtained using a diabatization scheme[52], following the procedures detailed in Supplementary Note 21. Normal-mode analyses of DPP–BDT and PDI fragments were used to compute the gradients of the excited-state PESs, providing the vibronic coupling constants that define the LVC Hamiltonian. Nuclear wave-packet propagations were conducted using the multi-layer (ML) extension of

MCTDH method, as implemented in the QUANTICS code[40], considering the vibrational modes that contribute most significantly to the total relaxation energy. The wavepackets were propagated for 200 fs with a time step of 0.5 fs, as shown in Supplementary Fig. 51. This approach captures the ultrafast population transfer between FE and CT states and the activation of characteristic DPP–BDT and PDI vibrational modes, as discussed in the main text.

### Data availability

The data underlying all figures in the main text are publicly available from the University of Cambridge repository https://doi.org/10.17863/CAM.126883 [Reference: Ghosh, P. et al. Data Supporting "Vibronically Assisted Sub-Cycle Charge Transfer at a Non-Fullerene Acceptor Heterojunction". Apollo −University of Cambridge Repository. https://doi.org/10.17863/CAM.126883 (2026)].

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

## Acknowledgements

The authors thank Professor Gregory D. Scholes (Princeton University), Dr. Christoph Schnedermann (University of Cambridge), Professor Richard H. Friend (University of Cambridge), and Professor Shahnawaz R Rather (University of Kentucky) for discussions. S.G. would also like to acknowledge Dr. Fabrizio Santoro (CNR-ICCOM) for useful discussions. This project has received funding from the European Research Council under the European Union's Horizon 2020 research and innovation program (Grant Agreement No. 758826 to A.R.). P.G. thanks the Cambridge Trust and the George and Lilian Schiff Foundation for a PhD scholarship and St John's College, Cambridge, for additional support and a title-A fellowship. H.B. acknowledges EPSRC (grant no EP/S003126/1). G.L. and S.G acknowledge the Italian Ministry of University and Research for funding provided by the European Union-NextGenerationEU-PNRR, Missione 4, Componente 2, Linea di investimento 1.2. The work in Mons received funding from the European Union's Horizon 2020 research and innovation program under grant agreement No. 964677, the Consortium des Équipements de Calcul Intensif (CÉCI), funded by the Fonds National de la Recherche Scientifique (F.R.S.-FNRS) under Grant No. 2.5020.11 as well as the Tier-1 supercomputer of the Fédération Wallonie-Bruxelles, infrastructure funded by the Walloon Region under Grant Agreement n1117545, and F.R.S.-FNRS. R.A. acknowledges support from St. John's

College, Cambridge. S.T.K. gratefully acknowledges funding from the European Union's Horizon 2020 Research and Innovation Programme under the Marie Skłodowska-Curie grant agreement no. 101022365. A.J.G. thanks the Leverhulme Trust for an Early Career Fellowship (ECF-2022-445), the Knut and Alice Wallenberg Foundation for a Wallenberg Academy Fellows award (KAW 2023.0082), and the Swedish Research Council (VR) for a Starting Grant (2024-03915).

## Author contributions

A.R., P.G., and H.B. conceived the project. P.G. developed, led the project, designed and built the experiments, and performed the resonant IVS and transient absorption spectroscopy measurements. J.R. synthesized the materials. G.L. and S.G. performed the quantum chemistry calculations. S.G. parametrized the LVC Hamiltonian and performed full quantum dynamics simulations. R.A. and P.G. performed the CW-Raman measurements. P.G. and A.J.G. performed the IR probed TA. S.T.K. and P.G. conducted the electrochemical measurements; S.J.Z. conducted the PDS measurements. A.R., D.B., and H.B. supervised the project. P.G., A.R., G.L., S.G., and D.B. cowrote the manuscript with input from all the coauthors.

## Competing interests

The authors declare no competing interests.
