## [Transparent Peer Review file · Nature Communications]

Vibronically Assisted Sub-Cycle Charge Transfer at a Non-Fullerene Acceptor Heterojunction

Corresponding Author: Professor Akshay Rao

Version 0:

Reviewer comments:

Reviewer #1

(Remarks to the Author)

The authors report studies of charge transfer process in donor-acceptor (DA) structures comprising a polymer labelled as Ref-P as donor and a non-fullerene-based acceptor, perylene diimide (PDI). PDI is covalently linked via a non-conjugated chain to the electron rich (TS-P2) and electron deficient (TS-P3) side of the polymer. The main idea of the paper is to demonstrate that ultrafast electron transfer times can also be reached in DA systems with small energy offsets. This contrasts the “common knowledge” that large DA energetic offsets are beneficial for an efficient electron transfer and may be beneficial for increasing the open-circuit voltage in photovoltaic devices.

The authors have used temporally short ~12 fs red and green pump pulses to perform transient absorption (TA) spectroscopy. For TS-P3, the TA spectra show femtosecond dynamics which have been assigned to ultrafast electron transfer from Ref-P to PDI. TA spectra recorded with reduced time resolution have been reported in a recent N. Chem publication by some of the authors. The TA measurements are complemented by careful sample characterization (as reported in the SI) and preliminary quantum chemical calculations. All in all, the work is detailed and thorough. Several questions arise that need to be addressed before a recommendation about publication of the manuscript can be made.

- 1) TS-P3 contains the PDI acceptor unit attached to electron deficient side of polymer. Under excitation with red pump pulses this results in fast TA dynamics (Fig. 2e) while the TS-P2, in which PDI is attached to the electron rich side, shows dynamics on a much slower picosecond time scale. Very little said about the dynamics in TS-P2. This should be explained in more detail.
- 2) It remains rather unclear how the electron transfer time of 15 fs is deduced from the data in Fig. 2e. It is not clear how the dynamics of the donor and acceptor population are deduced from these data. A global fit and, at least, a rate equation model to describe the time evolution of the spectra is necessary.
- 3) It is stated in the abstract that “...specific driving modes enable such CT rates by mixing single Frenkel exciton states CT state...”. I find it surprising that no evidence for such ultrafast vibronic couplings are seen in the TA spectra in Fig. 2e. Since the time resolution is claimed to be faster than any relevant vibrational mode in the system, I would expect that coherent wavepacket motion modulates the TA spectra as seen by other groups in TA spectroscopy. If the time resolution is indeed as high as claimed by the authors, the gradual evolution of the TA spectra with time seems to speak against the relevance of vibronic couplings. A much more in-depth discussion of the role of vibronic couplings on the TA spectra in Fig. 2e seems necessary.
- 4) The abstract claims “... unprecedented CT rates...”. This statement is incorrect, s., e.g., R. Huber et al., JPCB 106, 6494 (2002) and others.
- 5) No explanation has been given for microscopic origin of the observed fast electron transfer to non-covalently linked PDI electron acceptor. Is it through space to the stacked PDI moiety?
- 6) The assignment of 1283 cm⁻¹ observed in TS-P3 is confusing. The authors have assigned the 1283 cm⁻¹ mode to a C-C-stretching mode of the PDI radical anion since they observed a 1295 cm⁻¹ mode in the Raman spectra of an isolated PDI molecule. In the next paragraph they argue that this mode is an excited state mode of PDI because its frequency is 1283 and not 1295 cm⁻¹ as observed in isolated molecule. They support this argument by different control measurements in Fig. 2b. Honestly speaking, all these measurements seem to show (to me) pretty much the same lineshape of the Fourier transform band in the 1280 cm⁻¹ region. I do not think that a distinction between ground and excited state mode can be made on the basis of these measurements. For this, much more detailed control measurements are needed. Such a distinction can be achieved, for instance by analyzing the phase of the vibrational modulation of the TA spectra. A clear and convincing demonstration that the 1283 cm⁻¹ mode is an excited state vibration of PDI seems necessary to make the paper publishable.

7) The authors find evidence for quite rapid electron transfer dynamics and substantial couplings of the electronic excitations to high frequency modes. This strongly suggests that nonadiabatic dynamics resulting from vibronic couplings are important for the electron transfer rate and yield. Theoretical models that describe these dynamics should obviously take such nonadiabatic couplings and the effects of vibronic decoherence on the resulting quantum dynamics into account. When discussing their results, the authors speak of “ballistic electron transfer”, “conventional” field-driven coherence” and “reaction-driven coherences” and “impulsive forces” generating vibrational coherences. These are complicated concepts, partly borrowed from classical mechanics. These concepts are used without proper explanation, let alone definition, of the terms. This makes it very difficult for a reader to understand what the authors want to say. Even more, the use of such oversimplified, quasi-classical concepts may also be highly misleading. Theoretical models that can properly account for the effects of vibronic couplings on charge transfer dynamics in donor-acceptor systems are well developed (Tretiak, Prezhdov, Bredas, ...). Also, the decoherence phenomena can readily be described using, e.g., Redfield or Lindblad master equations. I strongly suggest to attempt such a modelling in order to support the conclusions that are drawn from the experiments.

Other minor comments:

There seems to be significant contribution of cross-phase modulations close to time zero especially seen from crosscuts extending over 100 fs. Therefore, it becomes important to show the subtraction scheme especially for analysing data in 20-40 fs waiting time.

Page 10, line 238, the reference to figure, should read Figure 3b?

Page 3, line 63, factors that govern

Reviewer #2

(Remarks to the Author)

The manuscript by Ghosh, et al. details the excited-state dynamics of model polymer/non-fullerene acceptor heterojunctions monitored with high-time resolution broadband transient absorption spectroscopy. The model systems are comprised of a dyketopyrrolopyrrole-benzodithiophene (DPP-DBT) polymer donor and a perylene diimide (PDI) acceptor in two different configurations: one with the PDI closely interacting with the electron-rich BDT unit (TS-P2), and the other where it is more closely associated with the electron-deficient BDP moiety (TS-P3). They observe sub-picosecond charge separation in TS-P3, while for TS-P2 they show charge separation in <15 fs, which is less than the vibrational period of the high-frequency Raman-active transitions available. The results are high quality and the analysis is sophisticated, and the understanding proposed will be of significant interest to the field. While the proposed explanation is compelling, there are numerous unanswered questions that need to be addressed before this work can be considered publishable in Nature Communications.

Major issues:

-The authors central claim is that charge transfer here is fast while still directly occurring from the local exciton state of the polymer and not an optically accessible charge-transfer state. They have performed photothermal deflection spectroscopy measurements and DFT calculations to rule out this possibility. However, the photothermal deflection spectroscopy was performed under quite different conditions (i.e. in a high polarity environment and in a solvent where aggregation is discouraged), and so this is not conclusive. The DFT calculations are also prone to functional- and basis set-specific artifacts that could miss such interactions. Generally, the UV/vis spectra usually give some indication of donor-acceptor interactions that give rise to new CT bands, but these data (Figure 1c) are not discussed in a meaningful way in this context. Specifically, there are substantial differences between the absorption spectra of TS-P2 and the constituent Ref-P/PDI spectra that are not discussed. The splitting and red-shifting of the low-energy polymer absorption band could easily indicate aggregation of the polymer backbone or charge-transfer interactions between the PDI and BDT. Similarly, though less drastically, the absorption of TS-P3 shows broadening of the vibronic progressions of both PDI and the polymer transitions. These differences in spectra lead to questions about aggregation and disorder that should be addressed: how rigid are these structures and what role does flexibility and overall disorder have on the dynamics? Are the polymer molecular weights (including those within the distribution) sufficient to allow additional interactions with the PDI unit? A discussion of the length of the polymer in the main text would add to the rigor of the discussion of the absorption spectrum. But moreover a better discussion of the UV/vis spectra and whether the clear differences in the spectra are indicative of ground state electronic interactions should be given.

-It is not clear how these results translate into solid-state design rules for better organic photovoltaic (OPV) device performance. Rapid charge separation from polymers to non-fullerene acceptors, including PDI is not novel and not inherently reliant on the mechanism proposed here. While this mechanistic approach will certainly be interesting to the community, it is not clear how these effects can be leveraged in devices. Better contextualization of these results is necessary to increase their impact and utilization by the community. Related, there are significant examples in Nature where fast, high yielding charge transfer occurs with minimal energy loss, such as the special pair in photosynthesis. This strategy has even been adapted in artificial systems for OPVs utilizing symmetry-breaking charge transfer [J. Am. Chem. Soc. 2015, 137, 16, 5397–5405]. A difference, of course, is that in those systems there are frequently indications of strong electronic coupling in the ground state, except in the case of null-aggregates which are known to exhibit excited-state symmetry breaking. In those and related cases, the role of vibrations has been investigated [J. Am. Chem. Soc. 2022, 144, 34, 15539–15548, Nat. Chem. 14, 786–793 (2022)], but this context is lacking here. As such, the notion that energy loss and charge transfer rates are “fundamental barriers” are overstatements.

-The ultrafast nature of the charge transfer in TS-P3 being driven by the 1512 cm⁻¹ polymer mode is intriguing, so the natural question is how general is this conclusion? The authors state that this mode is localized mostly in the DPP unit, which

seems consistent with the difference in dynamics between TS-P2 and TS-P3, however, this is not emphasized in the discussion. Indeed, it This is disappointing, as it may point towards the generality of this mechanism and its use in designing more efficient charge transfer systems. This element should be discussed more. Does the solvent play any specific role, either through inertial or dielectric stabilization or through coupling to the solute modes? Such effects would not be present in the solid-state device so this potential effect is worth clarification.

-The CW Raman spectra results for PDI where the 1283 cm⁻¹ mode is absent are compelling evidence for the lack of ground-state excitation of the PDI. However, the signal-to-noise ratio of these data is not particularly high and there may be indications of this missing feature close to the noise floor.

-Can the authors perform a short-time Fourier transform on the transient absorption data to see how the vibrational modes behave in time [J. Am. Chem. Soc. 2022, 144, 15539–15548]? This may better help demonstrate their proposed mechanism.

Minor issues:

-The abstract is very vague and should refer to the real types of systems being discussed in the text.

-Figure 2a: "S0Sn" absorption is presumably "S1Sn" absorption as stated on page 7.

-Figure 2b: What is the origin of the relatively slow rise in the Ref-P signal? This is also present to a lesser extent in the TS-P2 data.

-Figure 2d-f: It would be helpful to show a "zero-line" to guide the eye for when the transient absorption signals cross change sign.

-Figure 3b: what is the range of integration for the Fourier transform data?

-Page 9, line 214: "TP-P3" is presumably "TS-P3"

-Page 10/Supplementary Note 5: The agreement of the transient absorption spectrum with the spectrum of the chemically reduced PDI is not great. What is the origin of this disagreement? Aggregation is known to cause such broadening and shifting of anion spectra. Its use as supporting evidence is undermined by the lack of agreement and discussion.

-Page 21, the figure call in the methodology section is incomplete.

Version 1:

Reviewer comments:

Reviewer #1

(Remarks to the Author)

After the first round of reviewing, the manuscript has been improved substantially. The authors have responded to the comments and suggestions of both reviewers. They have added new measurements and new data analysis. This is very helpful and much appreciated.

In particular, I have enjoyed seeing the excited state quantum dynamics simulations which are now added as a new section in the Supplementary Information.

All in all, the extensive revisions made by the reviewers have supported the case of the authors quite strongly. I think that the manuscript is now much closer to being accepted for publication in Nature Communications.

I want to ask the authors to fully address the following points before recommending publication of this article.

a) I very much appreciate the details that are now given about the data analysis (SFig 41). I must say that I have my doubts about the claim ET time of 14.9 +/- 1 fs. I understand that the subtraction of coherent artefacts is challenging and I see that this has been done carefully. Yet, I think that red curve in 41b will also describe the data if an ET time of 20 or 25 fs is assumed (depending on how one takes into the account remaining "artefacts". I think that it is fair that the ET is fast (< 25 fs or so) but I do not think that the ET time can be extracted with one fs precision. I request to change this.

b) I guess that the time axis in SI Fig. 4 should be "ps" – not "fs" ;-)

c) The title in SFig. 49 should probably read "with/without linkers".

d) I acknowledge the discussion about Herzberg Teller couplings and agree qualitatively with what is said. Yet, I think that a much more careful analysis of the (possibly short-lived) excited state vibrations is needed before conclusions can be drawn. I would therefore suggest not to make too many strong statements about the origin of these vibrational coherences.

Since the authors have revised the manuscript very thoroughly, I gladly recommend publication of this article after these minor points have been carefully addressed.

Reviewer #2

(Remarks to the Author)

The authors have made significant efforts to address the reviewer comments, and the manuscript is much improved as a result. The manuscript is suitable for publication in Nature Communications.

Version 2:

Reviewer comments:

Reviewer #1

(Remarks to the Author)

The presentation of the material has been improved and I gladly support publication of the manuscript.

Response to the reviewers' comments

Reviewer #1 (Remarks to the Author):

The authors report studies of charge transfer process in donor-acceptor (DA) structures comprising a polymer labelled as Ref-P as donor and a non-fullerene-based acceptor, perylene diimide (PDI). PDI is covalently linked via a non-conjugated chain to the electron rich (TS-P2) and electron deficient (TS-P3) side of the polymer. The main idea of the paper is to demonstrate that ultrafast electron transfer times can also be reached in DA systems with small energy offsets. This contrasts the “common knowledge” that large DA energetic offsets are beneficial for an efficient electron transfer and may be beneficial for increasing the open-circuit voltage in photovoltaic devices. The authors have used temporally short ~12 fs red and green pump pulses to perform transient absorption (TA) spectroscopy. For TS-P3, the TA spectra show femtosecond dynamics which have been assigned to ultrafast electron transfer from Ref-P to PDI. TA spectra recorded with reduced time resolution have been reported in a recent N. Chem publication by some of the authors. The TA measurements are complemented by careful sample characterization (as reported in the SI) and preliminary quantum chemical calculations. All in all, the work is detailed and thorough. Several questions arise that need to be addressed before a recommendation about publication of the manuscript can be made.

Response: We sincerely thank the reviewer for their careful reading of our manuscript and their constructive feedback. We are particularly encouraged by the positive remarks that *“the TA measurements are complemented by careful sample characterization (as reported in the SI) and preliminary quantum chemical calculations. All in all, the work is detailed and thorough.”* The acknowledgement of the thoroughness of our measurements and analyses is highly motivating. In the following, we respond point-by-point to the reviewer’s questions.

1) *TS-P3 contains the PDI acceptor unit attached to electron deficient side of polymer. Under excitation with red pump pulses this results in fast TA dynamics (Fig. 2e) while the TS-P2, in which PDI is attached to the electron rich side, shows dynamics on a much*

slower picosecond time scale. Very little said about the dynamics in TS-P2. This should be explained in more detail.

Response: We agree with the reviewer that our earlier discussion of the TS-P2 dynamics required further elaboration. We have accordingly revised both the main text and the Supplementary Information.

As the reviewer correctly noted, in TS-P3 the PDI acceptor is attached to the electron-deficient DPP unit, whereas in TS-P2 it is linked to the electron-rich BDT unit. However in the charge transfer excited states for both the TS-P2 and TS-P3 the hole is more localised ($PR_h = 2$) on the DPP fragment. This electronic distribution previously established in the ref¹ leads to large separation of the electron (PDI) in the TS-P2. Consequently, charge transfer in TS-P2 proceeds via a longer-range, configuration-dependent pathway, resulting in slower kinetics.

The central question of this study is why charge transfer in TS-P3 occurs exceptionally rapidly for an organic system and whether vibrational coupling plays a decisive role. In this context, TS-P2 serves as a critical control system: its slower charge-transfer dynamics provide a benchmark for interpreting the vibrational features in TS-P3—particularly in assessing whether the 1283 cm^{-1} mode corresponds to an off-resonant ground-state wavepacket or to an wavepacket that is related to the charge-transfer excited state.

To quantify the charge-transfer rate under ‘red’ pump excitation, we analysed the TS-P2 transient absorption data after removing coherent artefacts (Supplementary Fig. 42). The differential trace is well described by a mono-exponential decay (red fit), yielding $\tau = 376$ fs—consistent with a slower, weakly coupled electron-transfer process with respect to the faster electron transfer in TS-P3 (sub-15 fs). This new analysis and the associated mechanistic explanation have been incorporated into the revised manuscript as detailed below.

Action:

Revised Main Manuscript:

In the section which describes the IR-probed 150fs resolved TA analysis is done:

“While TS-P2 exhibits electron transfer on a sub-picosecond timescale, TS-P3 undergoes rapid quenching of its neutral excitonic state almost concurrently with photoexcitation—within the instrument response (~ 150 fs)—rendering the process temporally unresolved. In both TS-P3 and TS-P2, the hole wave function of the lowest energy excited state is predominantly localized on the DPP segment¹, resulting in a greater spatial separation between the electron (localized on PDI) and hole for the case of TS-P2. This geometry in TS-P2 leads to weaker electronic coupling and consequently a slower, configuration-dependent CT process compared to the TS-P3 architecture.¹”

In the section which describes the ultrafast ‘red’ pump excitation:

“Turning to TS-P2, as shown in the Figure 2f, the dynamics of the spectral features at the very early time frames (20-40 fs) are silent like in Ref-P. The photogeneration of the PDI radical anion happens in much slower time scale (376 fs, SI fig. 42) in TS-P2 compared to TS-P3, Figure 2c. The slower electron transfer rate is in consistent with the previously described 150 fs IR-probed TA results where the TS-P3 dynamics remained unresolved.....”

Revised Supplementary Information

Supplementary Note 16. Extraction of the ultrafast charge-transfer dynamics in model heterojunction

Supplementary Fig. 42 | Difference trace of TS-P2 after removal of the coherent artefact fitted with a mono-exponential decay (red line), giving a time constant $\tau = 376$ fs.

2) It remains rather unclear how the electron transfer time of 15 fs is deduced from the data in Fig. 2e. It is not clear how the dynamics of the donor and acceptor population are deduced from these data. A global fit and, at least, a rate equation model to describe the time evolution of the spectra is necessary.

Response: We thank the reviewer for this insightful suggestion. Global analysis is indeed a rigorous and widely used method in time-resolved spectroscopy, enabling reliable extraction of kinetic and population-transfer information—provided the underlying system has well-separated, thermally populated states. However, in our present case, the earliest spectral dynamics (<50 fs) fall within the coherent regime: here, excitonic and charge-transfer (CT) states are strongly vibronically mixed and evolve as superposed wavepackets rather than independent populations. This rapid quantum coherence and coupling prevents the spectra from being represented as sums of time-independent basis functions, so conventional global fitting or rate-equation models cannot faithfully describe these ultrafast processes². To address this valuable comment, we have clarified our methodology and added new supporting analysis in the Supplementary Information (SI Fig. 41). In the revised manuscript, we directly extracted the differential kinetics of the donor and acceptor features to determine the characteristic ET timescale. As shown in Supplementary Fig. 41a, the raw $\Delta T/T$ traces of TS-P3 and Reference exhibit identical coherent artefacts around time zero (< 20 fs, vertical line in Fig. 41), but diverge immediately afterward (≥ 20 fs), with TS-P3 showing an additional ultrafast decay absent in the reference. Subtracting the reference trace isolates the CT-specific component (Supplementary Fig. 41b), which is well described by a mono-exponential decay with $\tau = 13.9 \pm 1$ fs. This value represents the experimentally resolvable upper limit of the ET process, approaching our 11 fs instrument response function, and therefore indicates that electron transfer completes within one vibrational period of the 1512 cm^{-1} DPP-localized mode. This interpretation is further supported by the 100 fs-resolved transient absorption kinetics (Fig. 2b), which track solvation and dielectric relaxation. In Ref-P, pronounced sub-ps solvation dynamics are observed, and these are partially retained in TS-P2 (the slower-ET analogue). In contrast, TS-P3 exhibits no such relaxation signature and instead decays directly into the product state immediately after photoexcitation. This

behaviour is consistent with a near-instantaneous charge transfer completed within one vibrational cycle of the 1512 cm^{-1} DPP-localized mode.

We believe that incorporating this analysis strengthens the manuscript and makes the interpretation clearer. To reflect the reviewer's suggestion, we have also added an explicit note in the main text stating that the sub-15 fs value corresponds to the *upper limit of the ET timescale derived directly from differential spectral kinetics*, rather than a fitted population-transfer rate. We thank the reviewer for prompting this addition, which significantly improves the clarity of the analysis.

Action:

Main Manuscript:

To quantify the ultrafast CT dynamics in TS-P3, kinetic analyses were performed after isolating the coherent artefact (Supplementary Fig. 41 and Note 17), which yielded a mono-exponential decay with $\tau = 14.9 \pm 1$ fs, establishing an upper limit for the electron-transfer process. Such sub-15 fs charge transfer is markedly faster than in conventional OPV systems, where CT times are typically well above 15fs, often by more than few order of magnitude³⁻⁶

Revised Supplementary Information

Supplementary Note 17: Extraction of the ultrafast charge-transfer dynamics in model heterojunction

To further substantiate the sub-15 fs charge transfer (CT) dynamics inferred from the transient absorption spectra (Fig. 2e, main text), we compared the time-domain kinetics of TS-P3 with those of the reference polymer (Ref-P), as shown in Supplementary Fig. 41. Panel a shows the averaged $\Delta T/T$ traces of TS-P3 (blue) and Ref-P (orange) at 790–820 nm, where the acceptor (PDI) absorption dominates⁷. Both datasets exhibit identical coherent artefacts around time zero (< 20 fs), arising from cross-phase modulation and pump–probe overlap. Beyond this temporal region, however, the TS-P3 trace displays a pronounced rapid decay that is completely absent in Ref-P, indicating an additional ultrafast process linked to charge separation. It is also important to note that beyond ~ 20

fs there is no further spectral evolution in Ref-P (Fig. 2d), confirming that the contribution of coherent artefacts or cross-phase modulation is minimal after this time (vertical dotted line in Supplementary Fig. 41a). To isolate this CT-specific contribution, the Reference trace was subtracted from the TS-P3 signal. The resulting differential kinetics (Supplementary Fig. 41b) were fit with a mono-exponential decay function, yielding a time constant of $\tau = 14.9 \pm 1$ fs. This value represents the experimentally resolvable upper limit of the CT timescale, approaching the 11 fs instrument response function. The result implies that electron transfer in TS-P3 completes essentially within a single vibrational period of the 1512 cm^{-1} DPP-localised mode. In the following Supplementary Note 21, we corroborate these experimental observations with quantum-dynamical calculations which show the presence of a DPP-localized mode at approximately 1500 cm^{-1} strongly coupled with the electron transfer.

Supplementary Fig. 41 | (a) Transient absorption kinetics ($\Delta T/T$) of TS-P3 (blue) and Ref-P (orange) averaged over 790–820 nm. The vertical dashed line marks time zero (< 20 fs), corresponding to the coherent artefact region. (b) Difference trace (TS-P3 – Ref-P) fitted with a mono-exponential decay (red line), giving a time constant $\tau = 14.9 \pm 1$ fs.

Such near-instantaneous transfer is consistent with the vibronically coherent regime discussed in the main text and is further supported by the 100 fs-resolved transient absorption dynamics (Fig. 2b, main text), where slower sub-picosecond solvation relaxation is evident in Ref-P and TS-P2 but absent in TS-P3. Together, these findings confirm that the < 15 fs component represents the coherent electron transfer event rather than subsequent solvent or structural relaxation.

3) It is stated in the abstract that "... specific driving modes enable such CT rates by mixing single Frenkel exciton states CT state ...". I find it surprising that no evidence for such ultrafast vibronic couplings are seen in the TA spectra in Fig. 2e. Since the time resolution is claimed to be faster than any relevant vibrational mode in the system, I would expect that coherent wavepacket motion modulates the TA spectra as seen by other groups in TA spectroscopy. If the time resolution is indeed as high as claimed by the authors, the gradual evolution of the TA spectra with time seems to speak against the relevance of vibronic couplings. A much more in-depth discussion of the role of vibronic couplings on the TA spectra in Fig. 2e seems necessary.

Response: We thank the reviewer for raising this important point. In many systems, vibronic coupling is indeed readily identified through clear oscillatory modulation of transient absorption features via a Franck–Condon–type detection mechanism. This results in periodic red and blue shift of the peak position. However it is noteworthy that under a Herzberg–Teller detection pathway^{8–10}, wavepacket motion primarily modulates the oscillator strength. As a result in such detection mechanism which is often found in organic molecules^{8–10}, oscillatory spectral blue and red shift is not present, which is a diagnostic feature of the nonc-Condon type Herzberg–Teller coupling^{8–10}. In such cases the vibrational coherence is mainly detected from the kinetic profile of individual probe wavelength.

Nevertheless, upon close inspection, the stimulated emission band of the polymer backbone in Fig. 2e exhibits oscillatory behaviour before it evolves into the PDI radical-anion absorption signature. To aid the reader, we have added a visual guide to highlight this feature. We also highlight the temporal oscillation in the time domain data in the figure 3a in the manuscript. We have also added the two-dimensional representation of the data where the oscillatory component of the spectral features are clear (Supplementary Fig. 9.2). In addition, we now include a dedicated Supplementary Section (Supplementary Note 7) with detailed discussion of vibronic coupling as the reviewer suggested, including appropriate citations due to word count constrain in the main manuscript. We believe this significantly improved the revised manuscript. We also highlight that our newly performed quantum dynamics simulations (Supplementary Note

21, revised SI) show that the DPP–BDT backbone chromophore is strongly coupled to several high-frequency vibrational modes (including the 1531 cm⁻¹ mode that drives ultrafast electron transfer) to the FE and CT excitons and the polymer cation. These results corroborate the experimental evidence of strong vibronic coupling observed under impulsive excitation.

Action:

Revised Supplementary Information

Supplementary Note 7: Additional discussion on the vibronic coupling of the model heterojunction

Here we discussed the vibronic coupling of model heterojunctions. In the electronic excited state (S₁) manifold, the vibrational ground state is represented as $\psi_{i=0}(x)$ and the first excited vibrational states are $\psi_{i=1}(x)$. After impulsive photoexcitation by a broadband laser pulse from the electronic ground state, multiple vibrational states will be populated, leading to, a newly generated non-stationary states which can be represented as:

$$\psi(r, t) = c_{i=0}(t)e^{-i\omega_0 t}\psi_{i=0}(r) + c_{i=1}(t)e^{-i\omega_1 t}\psi_{i=1}(r) + \dots$$

Where $c_{i=0}$, $c_{i=1}$, corresponds to the contribution of the each vibrational states to the non-eigen state.

The time-dependent molecular polarization can be described as -

$$P(t) = \langle \psi(r, t) | \mu | \psi(r, t) \rangle$$

Where μ is the dipole moment operator.

$$P(t)$$

$$= \langle c_{i=0}(t)e^{-i\omega_0 t}\psi_{i=0}(r) + c_{i=1}(t)e^{-i\omega_1 t}\psi_{i=1}(r) | \mu | c_{i=0}(t)e^{-i\omega_0 t}\psi_{i=0}(r) + c_{i=1}(t)e^{-i\omega_1 t}\psi_{i=1}(r) \rangle$$

$$P(t) = \mu_{01}(c_0^*c_1e^{-i(\omega_1-\omega_0)t} + c_1^*c_0e^{i(\omega_1-\omega_0)t})$$

The macroscopic polarization can be represented as $P_N(t) = N.P(t)$, where N is the number of molecules¹¹. Hence, the vibrational coherence generated by the superposition of two vibrational states can oscillate with $(\omega_1 - \omega_0) = \omega^{les}$. The oscillatory time

dependent change in the macroscopic polarization will vanish if the energetic bandwidth of the excitation laser source is lower than $\omega^{(es)}$.

Supplementary Fig. 9.1 | Two kind of resonant probing mechanism: a, Frank-Condon model and b, non-Condon Herzberg Teller model

Vibrational coherence can be detected through two complementary mechanisms. In the Franck–Condon (FC) picture (supplementary fig. 9.1a), nuclear motion periodically displaces the excited-state potential energy surface, leading to periodic red–blue shifts of electronic transitions in the transient absorption spectra. This results in a characteristic modulation of peak positions, which is readily observed when spectral features are well isolated. In contrast, in the Herzberg–Teller (HT) mechanism^{8–10}, vibrational motion modulates the electronic transition dipole moment or interstate coupling, producing oscillatory intensity changes in the transient features rather than pronounced spectral shifts. As a result of that if the vibrational coherence detection mechanism is not HT type mechanism then it is possible to detect the oscillatory spectral movement is possible.

Nuclear wavepacket motion in the vibration co-ordinate can be used as a probe for exciton-vibrational coupling¹². In such impulsive excitation, the amount of the oscillation is dependent on the electron–phonon coupling strengths are represented by a set of parameters $[\Delta, S, \lambda]$. Δ is a dimensionless displacement of the normal coordinate (as discussed above), S is the Huang–Rhys parameter, and λ is the reorganization energy.

Both S and λ exclusively dependent on the displacement (Δ) as follows:

$$S = \frac{\Delta^2}{2}; \quad \lambda = \hbar\omega S$$

where ω is the frequency of the optical phonon. λ ($\hbar\Delta\omega$) can be calculated from impulsive vibrational spectroscopy data using $A_{OSC} = \left(\frac{d OD}{d\omega}\right) \Delta\omega$, where OD is the optical density of the sample and A_{OSC} is the amplitude of the oscillations. A_{OSC} of any vibrational modes can be obtained by fitting the residuals to a damped sine function. Relative A_{OSC} between different modes for comparison purposes, can be obtained from the relative FFT amplitudes.

The wavelength-resolved transient absorption dynamics of Ref-P and TS-P3 are shown in Supplementary Fig. 9.2. These data display clear oscillatory profiles, consistent with exciton–vibrational coupling. The maps also highlight distinct excited-state behaviours shaped by polymer architecture, with TS-P3 exhibiting modified spectral features on the ultrafast timescale that can be attributed to the presence of the through-space PDI units.

Supplementary Fig. 9.2 | Wavelength-resolved transient absorption dynamics of Ref-P and TS-P3 polymers: a, Transient absorption map of the Ref-P polymer, showing the temporal evolution of its photoexcited states. **b,** Transient absorption map of the TS-P3 space polymer, highlighting the influence of through-space PDI units on the excited-state dynamics. **c,d,** corresponding zoomed-in versions Ref-P(c) and TS-P3 (d).

Effective Huang–Rhys factor from band-integrated IVS: The high-frequency IVS contribution ($1000\text{--}1500\text{ cm}^{-1}$) was isolated by band-pass filtering the TA residuals after background subtraction as shown in supplementary figure 9.3. An effective modulation depth m_{band} was defined as the RMS amplitude of the band-passed residual divided by the smooth TA background at the same probe wavelength. For Herzberg–Teller detection we used $S_{\text{eff}} \approx m_{\text{band}}$; for Franck–Condon detection we used $S_{\text{eff}} \approx m_{\text{band}} \bar{\Gamma} / (2\hbar\bar{\omega})$, where $\bar{\Gamma}$ is the representative half-width and $\bar{\omega}$ the band center. Measured amplitudes were corrected for the 12 fs Gaussian IRF using $H(\omega) = \exp[-\omega^2\sigma^2/2]$ ($\sigma = 5.1\text{ fs}$). For a quick

mode-resolved estimate, FFT peak areas within $\pm 10 \text{ cm}^{-1}$ of each mode were used to apportion S_{eff} into S_i via weights $w_i \propto \text{FFT area}$, followed by mode-specific IRF corrections. Reported S values thus represent conservative (band-integrated) lower bounds on the high-frequency reorganization. From this analysis method we estimate effective S_i for high freq. modes (1000-2000 cm^{-1}) is 0.52 and for the 1515 (± 10) cm^{-1} mode the S_{eff} is 0.08 which is comparable to the previously reported BDT-DPP polymers^{13,14}.

To connect the vibronic coupling observed in the ultrafast photophysics, we carried out quantum dynamics simulations using the LVC Hamiltonian for the DPP-BDT and PDI fragments detailed in the supplementary note 21. This approach allows us to track in real time how nuclear motion along specific normal modes influences the evolution of the FE and CT states and the formation of the charge-separated configurations. By explicitly propagating the coupled electron-nuclear wavepacket, the simulations reveal the vibrational modes that most efficiently drive population transfer between these manifolds (supplementary fig. 51). These simulations clearly show that several high-frequency modes of the DPP-BDT are strongly coupled with both the exciton as well as the cationic states (supplementary fig. 50).

Supplementary Fig. 9.3 | Ultrafast kinetic traces, residuals, and corresponding Fourier transforms: a, Measured transient absorption kinetics (green) and the corresponding moving-average background (red, 200 fs window). **b**, Residuals obtained after subtracting the exponential kinetic fit (orange, top) and band-pass fit (blue, bottom), revealing coherent oscillations associated with vibrational modes. **c**, Fourier transforms (FFT) of the respective residuals, showing dominant vibrational modes in the 400–1600 cm^{-1} range. **d**, Comparison between the raw FFT spectrum (orange) and the time-resolution-corrected FFT (grey dashed), highlighting sharpening of vibrational peaks after correction.

Main manuscript

(Revised) figure 2e

4) The abstract claims “... unprecedented CT rates...”. This statement is incorrect, s., e.g., R. Huber et al., *JPCB* 106, 6494 (2002) and others.

Response: We thank the reviewer for pointing this out. We recognise the pioneering work of Huber and co-workers, for example:

- Huber, R. et al., *J. Phys. Chem. B* **106**, 6494–6499 (2002).
- Huber, R. et al., *Chem. Phys.* **285**, 39–45 (2002).

The study reported sub-10 fs electron transfer times in dye/semiconductor oxide colloidal systems. We agree that these timescales are comparable to the sub-cycle charge transfer observed in our work. However, as the reviewer notes, those hybrid organic–inorganic systems benefit from large dielectric constants that facilitate free charge formation. By contrast, our system is a **purely organic donor–acceptor heterojunction**, in which such sub-15 fs ultrafast transfer is underexplored to the best of our knowledge.

Action: In recognition of this point, we have revised the abstract to replace “unprecedented CT rates” with “ultrafast CT rates *in purely organic systems.*”

5) *No explanation has been given for microscopic origin of the observed fast electron transfer to non-covalently linked PDI electron acceptor. Is it through space to the stacked PDI moiety?*

Response: We thank the reviewer for raising this point. Because the PDI is connected via a non-conjugated tether, electron transfer cannot proceed through a ‘through-bond’ pathway but must occur via a ‘through-space’ mechanism in both TS-P2 and TS-P3, as the reviewer suggested. This distinguishes our constructs from previously reported model heterojunctions such as polymers with pendant donor/acceptor groups¹⁵⁻¹⁸ or block copolymers¹⁹⁻²¹, where through-bond interactions dominate.

To further probe the microscopic origin of CT, we also examined a physical mixture of Ref-P and PDI at the same overall composition. As shown in the SI (originally included for a different purpose), this mixture does not display the PDI radical-anion signatures or accelerated bleach decay that are observed in TS-P3. Instead, its kinetics closely resemble Ref-P alone, consistent with negligible ET. This control confirms that stochastic encounters between Ref-P and PDI do not yield efficient femtosecond CT, and that the pre-organised geometry enforced by the covalent tether is essential for enabling through-space coupling.

To address the reviewer’s question regarding whether the donor–acceptor interaction in TS-P3 is mediated through the non-conjugated linkers or arises from direct through-space coupling, we performed additional quantum chemical calculations using a diabaticization approach on two model systems. These include (a) the full TS-P3 dimer, which retains the alkyl linkers connecting the DPP–BDT unit to the PDI, and (b) a truncated version in which these linkers were removed (supplementary fig.). As detailed in supplementary note 21, both models yield essentially identical electronic couplings, and the corresponding diabatic Hamiltonians are reported in supplementary table 5. This demonstrates that the linkers do not play a role in the coupling mechanism and confirms that the donor–acceptor interaction is dominated by through-space, rather than through-bond, coupling.

The key difference between the two architectures then lies in the site of attachment:

– **TS-P3:** PDI is placed adjacent to the DPP moiety, where close co-facial proximity combines with strong vibronic coupling to the $\sim 1512\text{ cm}^{-1}$ DPP-localised mode.

– **TS-P2:** PDI is attached near the BDT unit, where the through-space interaction is weaker and no equivalent vibrational driving mode is engaged, resulting in slower $\sim 376\text{ fs}$ transfer.

Action:

Main text:

“Because the PDI is attached via a non-conjugated linker, CT in TS-P3 occurs through space. This is supported by the absence of CT in a Ref-P + PDI mixture (Supplementary Note 16) and by electronic coupling calculations performed on TS-P3 with and without the linkers (Supplementary Note 21), which produce nearly identical FE-CT couplings (supplementary table 5), confirming that linkers do not mediate the interaction. Thus, the ultrafast $<15\text{ fs}$ CT arises uniquely from the tethered through-space geometry and its coupling to the DPP-localised vibrational mode.”

Supplementary Information

Supplementary Note 16: Microscopic Origin of Charge Transfer

Because the PDI is connected to the polymer via a non-conjugated alkyl tether, charge transfer cannot proceed through a through-bond pathway but should occur via through-space interactions between the polymer donor backbone and the PDI acceptor. This design distinguishes our ‘through space’ constructs from other model heterojunctions, such as polymers with pendant donor/acceptor^{15–18} groups or block copolymers^{19–21}, where through-bond coupling is significant.

Additional evidence for the ‘structurally predefined through-space’ nature of the transfer comes from a control experiment in which Ref-P and PDI were physically mixed at the same overall composition. As shown in the supplementary figure 20, this mixture does not exhibit the characteristic PDI radical-anion signatures or accelerated polymer bleach

decay observed in TS-P3. Instead, its kinetics closely resemble Ref-P alone, indicating negligible charge transfer. This confirms that random intermolecular encounters do not support efficient femtosecond CT, and that the covalent tether is essential for enforcing donor–acceptor ‘through-space’ proximity and dynamic CT-LE mixing through the vibrational modes.

To evaluate whether donor–acceptor coupling in TS-P3 is transmitted through the alkyl linkers or arises from through-space interactions, we conducted diabaticization calculations on two variants of the model system: (a) the full TS-P3 dimer with all non-conjugated linkers included, and (b) a truncated structure with these linkers removed (supplementary fig. 49). The results of these quantum-dynamics-based calculations are presented in supplementary note 21, and the corresponding diabatic Hamiltonians are compiled in supplementary table 5. In both cases, the extracted electronic couplings are effectively identical, indicating that the linkers do not contribute to the coupling pathway. These findings confirm that the donor–acceptor interaction in TS-P3 is governed by through-space rather than through-bond coupling.

The difference between TS-P2 and TS-P3 therefore lies in the attachment site:

- TS-P3: PDI lies adjacent to the electron-deficient DPP moiety, enabling close co-facial geometry. Strong vibronic coupling to the $\sim 1512\text{ cm}^{-1}$ which is a DPP-localised vibrational mode dynamically mix singlet (Frenkel exciton, FE) and CT states following photoexcitation, allowing for CT to be achieved on timescales much faster than would be expected when considering purely static electronic coupling interactions.
- TS-P2: PDI is tethered near the electron-rich BDT moiety, where spatial overlap and vibronic driving are weaker, leading to a slower $\sim 376\text{ fs}$ transfer.

Together, these results show that through-space coupling is the operative mechanism in both systems, but vibronic enhancement at the DPP site makes CT in TS-P3 exceptionally fast.

Supplementary Fig. 20 | Resonant impulsive vibrational spectroscopy on the chemical mixture of the Ref-P and PDI maintaining the same number of PDI content per BDT-DPP unit as TS-P3 (excitation by the 700 nm centred broadband pulse) : a, Transient absorption ($\Delta T/T$) map. b, Transient absorption spectra ($\Delta T/T$) plotted against probe wavelength (nm) for different pump-probe delay. c, Transient absorption kinetics extracted from the 790-830 nm. d, Probe wavelength resolved IVS map. The red dotted line highlights the position of the 1283 cm^{-1} .

** We have not reproduced the supplementary table 5, supplementary note 21 and supplementary figure 49 here to avoid repeat as that is included in the action of the point number 7 made by the reviewer.

6) The assignment of 1283 cm^{-1} observed in TS-P3 is confusing. The authors have assigned the 1283 cm^{-1} mode to a C-C-stretching mode of the PDI radical anion since they observed a 1295 cm^{-1} mode in the Raman spectra of an isolated PDI molecule. In the next paragraph they argue that this mode is an excited state mode of PDI because its frequency is 1283 and not 1295 cm^{-1} as observed in isolated molecule. They support this

argument by different control measurements in Fig. 2b. Honestly speaking, all these measurements seem to show (to me) pretty much the same lineshape of the Fourier transform band in the 1280 cm⁻¹ region. I do not think that a distinction between ground and excited state mode can be made on the basis of these measurements. For this, much more detailed control measurements are needed. Such a distinction can be achieved, for instance by analyzing the phase of the vibrational modulation of the TA spectra. A clear and convincing demonstration that the 1283 cm⁻¹ mode is an excited state vibration of PDI seems necessary to make the paper publishable.

Response:

We thank the reviewer for raising this important point and acknowledge that our original description was not sufficiently clear. The observation of a 1295 cm⁻¹ mode in the cw-Raman spectrum of isolated PDI was used to identify the 1283 cm⁻¹ vibrational coherence as structurally originating from the PDI moiety, since the polymer backbone has no mode in the 1275–1325 cm⁻¹ region. However, as the reviewer rightly points out, the small frequency shift alone cannot serve as definitive evidence that the 1283 cm⁻¹ coherence arises from the excited state, and we have revised the manuscript to clarify this.

We agree that in the full-range stacked comparison, the Fourier transform bands in the 1280–1300 cm⁻¹ region appear broadly similar, which may have given the impression that no clear distinction can be made. To address this, we now provide a zoomed-in, normalised, and overlapped comparison in the 1280–1300 cm⁻¹ window in the revised SI (Supplementary Fig. 43), which highlights reproducible differences between the 1295 cm⁻¹ ground-state Raman/IVS modes of isolated PDI and the 1283 cm⁻¹ feature observed in TS-P3. The 1291 cm⁻¹ ground-state vibrational coherence and the 1295 cm⁻¹ cw-Raman mode both show symmetric lineshapes, with the latter displaying a slight high-energy shoulder. In contrast, the 1283 cm⁻¹ feature in TS-P3 is clearly shifted, broadened, and asymmetric. All Fourier transforms were performed using the same time-domain window (40–1700 fs) to ensure an unbiased comparison of lineshapes.

We agree with the reviewer that early-time phase analysis of oscillations can provide a diagnostic to distinguish between ground- and excited-state coherences. In principle,

cosine-like wavepackets (ISRS in the ground state) have phase $m\pi$, whereas sine-like wavepackets (excited-state launch) have phase $(2m+1)\pi/2$, requiring precision on the order of $\tau/4$ to discriminate. For the 1283 cm^{-1} mode ($\tau \approx 26\text{ fs}$, $\tau/4 \approx 6.5\text{ fs}$), this distinction is technically challenging: the quarter-period is comparable to the 11 fs IRF, and most importantly the early-time window is further convoluted by coherent artefacts, cross-phase modulation, and other polymer backbone Franck–Condon modes (Fig. 3a), all of which bias phase retrieval. For this reason, phase analysis is applied to slower modes (100–500 fs period), as in the 106 fs mode reported by Rather *et al.* (Nat. Chem. 2021²²) or 416 fs mode reported by Kong *et al.* (Chem. Sci, 2024²³). Moreover, the analysis of our results shows that the 1283 cm^{-1} coherence is not launched directly by the light field but is instead driven by the ultrafast charge transfer. In such cases, theory and simulations predict additional phase behaviour: a $\pi/2$ shift relative to the reactant coherence (which could superficially resemble a ground-state ISRS signature) and a π shift relative to other Franck–Condon active modes. This scenario is consistent with the behaviour described by Rather *et al.* (Nat. Chem. 2021²²). However, such analyses have only been demonstrated for lower-frequency modes with long periods, and are not technically reliable for the present high-frequency regime.

As the early time few cycle phase analysis is challenging for this particular case, we provide two conclusive evidence that we have revised in the main and SI that shows the mode is not a ground state mode. If it was a ground state mode it would be generated by the ISRS mechanism (often referred as off-resonant IVS²⁴) which we now highlight properly in the revised manuscript.

- We analysed the spectral dependence (figure 3c, main) of the vibrational coherence of the 1283 cm^{-1} mode. A more important diagnostic is that the 1283 cm^{-1} coherence in TS-P3 is spectrally localised to the absorption window of the PDI radical anion (650–850 nm), and not to the neutral PDI bleach-tail nor having a flat spectral profile throughout. This indicates that the oscillation is resonantly detected through the excited-state CT absorption, rather than through ground-state Raman scattering. We therefore assign the 1283 cm^{-1} coherence to the PDI radical anion which is formed during ultrafast charge transfer. Such deep-red (650-850 nm) spectral dependence can not be explained if the vibrational

coherence is generated on the ground state. We have revised Fig. 3c to include a comparison between the PDI radical-anion feature extracted from the TA data along with the chemically reduced PDI radical-anion absorption, plotted with the spectral dependence of the 1283 cm^{-1} mode

- We also note that TS-P2 and a physical mixture of Ref-P and PDI, both of which contain the same amount of PDI, do not show the 1283 cm^{-1} feature. This absence indicates that the mode is not generated on the ground state of PDI via an ISRS mechanism, but is instead associated with the radical anion formed in TS-P3 through fast electron transfer reaction.

Now in the revised main text we have highlighted this two point to establish the vibrational coherence at 1283 cm^{-1} is generated not from ground state ISRS mechanism. To further clarify this assignment, we have now included a decision-tree summary in the Supplementary Information (SI Fig. 44) that collates all experimental and computational evidence.

We would like to highlight that in light with this comment, we computed the mode-specific gradients λ_k^α that define the linear vibronic coupling (LVC) Hamiltonian using the normal modes of isolated DPP-BDT and PDI fragments at the TS-P3 monomer geometry (as detailed in supplementary note 21). Ground-state gradients were obtained from the harmonic QM Hessian, and excited-state gradients for the locally excited, cationic, and anionic states were constructed within the Vertical Gradient (VG) approximation as implemented in FCclasses3.0. This approach reveals that several high-frequency DPP-BDT modes exhibit large displacements, with the 1531 cm^{-1} mode acting as the dominant driving coordinate. The PDI modes, in contrast, begin to evolve only after the CT states are populated via this fast electron transfer, in agreement with the experimental observations (supplementary figure 51). This is especially clear for the PDI mode at 1299 cm^{-1} , whose frequency coincides with what has been found experimentally (corresponding to experimental frequency 1283 cm^{-1}). In our impulsive vibrational spectroscopy measurements, the vibrational coherence along the 1283 cm^{-1} coordinate appears only after electron transfer, fully consistent with the simulations, which identify this mode as a PDI-anion vibration activated following CT-state formation.

Action:

Revised main manuscript excerpt

Page-10

We can assign this newfound ω^β vibrational coherence to a high-frequency C-C stretching mode of the PDI acceptor moiety, since we observe a mode at similar frequency regime for an isolated PDI molecule with CW Raman spectroscopy (1295 cm^{-1} , blue spectra, Figure 3b(4)) and off-resonant IVS (1291 cm^{-1} , grey spectra, Figure 3b(4)). In addition, the polymer backbone chromophore has no mode in this region 1275–1325 cm^{-1} further confirmed by vibronic simulations with displaced harmonic oscillator model (supplementary fig. 50).

Page-11

First, we rule out the possibility that ω^β wavepacket is an off-resonant wavepacket launched on the ground state surface of PDI, i.e. the possibility that the ultrashort optical pulse, off-resonant with the electronic transition of PDI.....In contrast, the vibrational coherence at 1283 cm^{-1} (ω^β), obtained by resonant impulsive excitation of the polymer backbone in TS-P3, shows its strongest amplitude at probe wavelengths between 650 and 850 nm (black trace, Fig. 3c, top). This spectral envelope closely matches the PDI radical-anion feature retrieved from the transient absorption data (red trace, Fig. 3c, top), and is further qualitatively complemented by the absorption profile of the chemically reduced PDI⁻ species (blue trace, Fig. 3c, top).

Page-13

“See Supplementary Fig. 44 for a decision-tree summary of the evidence supporting the assignment of the 1283 cm^{-1} coherence.”

Page-14

To further explore these experimental observations, we constructed a linear vibronic coupling model for a representative TS-P3 monomeric unit, including the most relevant vibrational modes that couple to the localized (diabatic) FE state on the DPP–BDT unit as well as to two CT states (in which the electron is transferred from DPP–BDT to PDI).

Details are provided in methods and supplementary note 21. Using this model, we performed full quantum dynamics simulations with the multi-layer multi-configurational time-dependent Hartree (ML-MCTDH) method²⁵. The simulations reveal that the PDI vibrational mode at 1299 cm⁻¹—coincident with the experimentally observed ω^β feature in Fig. 3b—begins to evolve only after the CT states become populated through ultrafast electron transfer (see Supplementary Fig. 51). In contrast, the DPP-BDT polymer modes start oscillating immediately at time zero and act as driving modes for the CT as we discuss below. We note in passing that several groups have attempted similar excited state dynamics simulations, using not only full quantum dynamics approaches but also mixed quantum–classical molecular dynamics^{26–28}.

Revised supplementary information excerpt

Supplementary Note 18. Further discussion on the assignment of the 1283 cm⁻¹ Vibrational Coherence in TS-P3

The assignment of the 1283 cm⁻¹ vibrational coherence (VC) observed in TS-P3 requires careful consideration, as it lies close to the 1295 cm⁻¹ Raman-active mode of isolated PDI. CW-Raman and off-resonant impulsive vibrational spectroscopy (IVS) of isolated PDI show a 1295 cm⁻¹ mode (and a 1291 cm⁻¹ coherence in IVS), both with symmetric lineshapes (Supplementary Fig. 43). By contrast, the 1283 cm⁻¹ feature in TS-P3 is shifted, broadened, and asymmetric. The polymer backbone has no modes in the 1275–1325 cm⁻¹ region, confirming that this feature originates from the PDI moiety. Below, we collate experimental, spectroscopic, and theoretical evidence to demonstrate that this feature arises from the PDI radical anion generated during ultrafast charge transfer (CT), rather than from a ground-state impulsive Raman (ISRS) response.

- The 1283 cm⁻¹ coherence is tightly confined to the spectral region corresponding to the PDI radical-anion absorption (650–850 nm; Fig. 3c) and is absent from the neutral PDI bleach tail. This spectral localisation shows that the oscillation is resonantly detected through the excited-state charge-transfer (CT) absorption,

rather than arising from ground-state Raman pathways. In particular, the pronounced deep-red localisation is inconsistent with ground-state ISRS, which would produce a broadband response extending across the neutral bleach. We have **revised** the figure 3c to incorporate the absorption of the PDI radical anion feature retrieved from TA for the comparison.

- TS-P2, which contains the same PDI concentration as TS-P3, shows no 1283 cm^{-1} coherence, thereby ruling out any assignment to vibrational coherence generated by impulsive Raman scattering in the ground-state manifold. (Figure 3b(2), main)
- A physical mixture of Ref-P and PDI with the same composition as TS-P3 likewise shows no 123 cm^{-1} coherence, and its kinetics closely match those of Ref-P alone. This further rules out an origin in vibrational coherence generated by impulsive Raman scattering in the ground-state manifold of TS-P3. (Supplementary Fig. 20)

To aid interpretation, we provide a decision-tree schematic (Supplementary Fig. 44) that summarises the combined evidence. Together, these results confirm that the 1283 cm^{-1} coherence in TS-P3 arises from the PDI radical anion on the excited-state CT surface, and not from ground-state ISRS.

Supplementary Fig. 43 | Fourier transform spectra of vibrational coherences. Comparison of the resonant IVS spectrum of TS-P3 (orange) with cw-off-resonant Raman (blue) and off-resonant IVS (gray) spectra of isolated PDI. While the $1291\text{--}1295\text{ cm}^{-1}$ ground-state Raman/IVS modes of PDI exhibit symmetric lineshapes, the 1283 cm^{-1} feature in TS-P3 is distinctly shifted, broadened, and asymmetric. All Fourier transforms were performed over the same time window (40–1700 fs) to enable unbiased comparison.

Supplementary Fig. 44 | Evidence for assignment of the 1283 cm⁻¹ vibrational coherence in TS-P3. Decision tree summarising experimental and computational evidence for the microscopic origin of the 1283 cm⁻¹ mode.

To quantify vibronic interactions in TS-P3, we computed the mode-specific gradients λ_k^α that define the linear vibronic coupling (LVC) Hamiltonian in Eq. S21.4 as explained more in detail in Supplementary Note 21, using the normal modes of isolated DPP-BDT and PDI fragments evaluated at the TS-P3 monomer geometry. Ground-state gradients were obtained from the harmonic QM Hessian, while excited-state gradients for the locally excited, cationic, and anionic states were constructed within the Vertical Gradient (VG) approximation (see supplementary note 21). This framework reveals that several high-

frequency DPP–BDT modes exhibit large displacements, with the 1531 cm^{-1} mode acting as the principal driving coordinate in the reduced model. The PDI modes, in contrast, begin to evolve only after the CT states are populated via this fast electron transfer, in agreement with the experimental observations (supplementary figure 51). This is especially clear for the PDI mode at 1299 cm^{-1} , whose frequency coincides with what has been found experimentally. In our impulsive vibrational spectroscopy measurements, the vibrational coherence along the 1283 cm^{-1} coordinate appears only after electron transfer, fully consistent with the simulations, which identify this mode as a PDI-anion vibration activated following CT-state formation.

7) The authors find evidence for quite rapid electron transfer dynamics and substantial couplings of the electronic excitations to high frequency modes. This strongly suggests that nonadiabatic dynamics resulting from vibronic couplings are important for the electron transfer rate and yield. Theoretical models that describe these dynamics should obviously take such nonadiabatic couplings and the effects of vibronic decoherence on the resulting quantum dynamics into account. When discussing their results, the authors speak of “ballistic electron transfer”, “conventional field-driven coherence” and “reaction-driven coherences” and “impulsive forces” generating vibrational coherences. These are complicated concepts, partly borrowed from classical mechanics. These concepts are used without proper explanation, let alone definition, of the terms. This makes it very difficult for a reader to understand what the authors want to say. Even more, the use of such oversimplified, quasi-classical concepts may also be highly misleading. Theoretical models that can properly account for the effects of vibronic couplings on charge transfer dynamics in donor-acceptor systems are well developed (Tretiak, Prezhdo, Bredas, ...). Also, the decoherence phenomena can readily be described using, e.g., Redfield or Lindblad master equations. I strongly suggest to attempt such a modelling in order to support the conclusions that are drawn from the experiments.

Response:

We thank the reviewer for raising this important point about simulating the dynamics of the ultrafast charge separation process in TS-P3 as observed in our experiments. Following the reviewer's suggestion, we performed additional simulations. In particular, (i) we confirmed the reviewer expectation of strong vibronic couplings of the exciton and related CT states with a number of modes localized both on DPP-BDT polymer (see for Table S6 and Fig. S51); (ii) we built a suitable ab-initio parametrized Linear Vibronic Coupling (LVC) Hamiltonian representing the TS-P3 systems in order to perform full quantum dynamics using the gold-standard method for such a task, that is Multi-Configurational Time Dependent Hartree (MCTDH)²⁹⁻³³.

We have described the set-up of the model Hamiltonian and the outcome of our Quantum Dynamics (QD) simulations in a new Supplementary note in the SI titled: Excited-state quantum dynamics. There, we first explain how we have built the LVC Hamiltonian. For the electronic part of the Hamiltonian, we have computed the lowest diabatic states of a representative TS-P3 monomer unit that shows a Frenkel exciton state on the DPP-BDT unit and two lower energy CT (DPP-BDT →PDI) states where the electron travels from DPP-BDT to the PDI. For the vibrational part we included the most relevant modes for the dynamics among which the high frequency driving mode (reported already in the main text), which couples both to the exciton and the cation of the polymer, and the modes of the PDI. Our simulations nicely show that the modes localized on the PDI start moving following the ultrafast population transfer from the DPP-BDT to the PDI.

The results show an ultrafast population transfer from the DPP-BDT occurring in approximately ~75 fs when the energy off-set between LE and CT states is set to about 200 meV which is consistent with what we found from our photoluminescence⁷ and ER-EIS measurements (supplementary note 1). We note our simulated timescales are still slower than those observed experimentally, and the reason might be related to several simplification that are necessary to make the QD feasible on such a complex system, as for instance: the approximation of using the same gradients for different CT states, the reduced number of nuclear degrees of freedom and the absence of non-local electron-vibrational couplings. Despite these shortcomings we believe our new simulations provided additional important insights as we explain in the text below helpful to further support our experimental data.

As mentioned by the reviewer, a number of groups have indeed attempted similar simulations using either full quantum^{26,27} as well as mixed quantum-classical molecular dynamics^{26,28} and we have acknowledged this amount of work in our revision by citing a few of these works on page 14.

We thank the reviewer for highlighting the need for clearer contextualisation of technical concepts. We have now revised the manuscript accordingly. Guided by our new simulation results, we have expanded the explanations of these concepts and improved the transitions surrounding their introduction. These changes appear on pages 14, and 19, where we now define the terms more explicitly and cite the relevant literature.

Action: The action is the creation of an additional Supplementary note as explained below.

Supplementary Note 21: Excited-state Quantum dynamics

Linear Vibronic Coupling Hamiltonian for TS-P3

The linear vibronic coupling Hamiltonian used to describe the ultrafast dynamics of the TS-P3 polymer is written as a sum of electronic, vibrational and electron-vibronic Hamiltonians:

$$\hat{H} = \hat{H}^{el} + \hat{H}^{vib} + \hat{H}^{el-vib} \quad (\text{S21.1})$$

The electronic Hamiltonian (\hat{H}^{el}) describes the various electronic excitations of the system in the diabatic representation. These include localized excitations (FE) on the DPP–BDT polymer backbone and charge-transfer (CT) excitations between the DPP–BDT polymer and the PDI unit. More specifically,

$$\hat{H}^{el} = \hat{H}_{\text{FE}} + \hat{H}_{\text{LE-CT}} \quad (\text{S21.2})$$

Here, the diagonal elements correspond to the vertical energies of the diabatic excitations, while the off-diagonal elements, V_{kl} , represent the excitonic couplings that describe interactions between the FE and CT states included in the model. The electronic Hamiltonian is constructed using a diabaticization procedure starting from the adiabatic states of the TS-P3 polymer, as detailed in the next paragraph.

The kinetic and potential components of the vibrational Hamiltonian \hat{H}^{vib} are written in terms of the M (mass-weighted) dimensionless coordinates q_α and their conjugate momenta p_α of the modes of the TS-P3 system represented using:

$$\hat{H}^{vib} = \hat{K}^{vib} + \hat{V}^{vib} = \frac{1}{2} \sum_{\alpha} \omega_{\alpha} p_{\alpha}^2 + \frac{1}{2} \sum_{\alpha} \omega_{\alpha} q_{\alpha}^2 \quad (\text{S21.3})$$

Where ω_{α} is the frequency of the specific hierarchically selected mode α (considering $\hbar = 1$). Note that a dimensionless coordinate q_{α} and related momenta p_{α} are found from the analogue quantity with dimensions by: $q_{\alpha} = \sqrt{\frac{\omega_{\alpha}}{\hbar}} \sqrt{m} x_{\alpha}$ and $p_{\alpha} = \frac{m v_{\alpha}}{\sqrt{m \omega \hbar}}$, respectively.

The electronic excitations, i.e., the formation of FE states and CT states, are strongly coupled to high-frequency vibrational modes as shown also by our impulsive vibrational spectroscopy experiments. Such electron-vibrational interactions are presented by the following Hamiltonian:

$$\hat{H}^{el-vib} = \hat{H}^{FE-vib} + \hat{H}^{CT-vib} \quad (\text{S21.4})$$

The coupling between these quantized vibrational modes and the LE and CT excitations are incorporated into our Hamiltonian Eq. S21.1 by using a quantum displaced harmonic oscillator as explained the next paragraph and done by other groups.^{34,35}

The first term represents the coupling between FE states (for instance those on the TS polymer) and a given mode α is:

$$\hat{H}^{FE-vib} = \sum_{k=1}^{N_{LE}} \sum_{\alpha}^{N_{exc}} \lambda_k^{\alpha, LE} q_{\alpha} |FE_k\rangle \langle FE_k| \quad (\text{S21.5})$$

Similarly, the coupling between CT states and the nuclear degrees of freedom is given by:

$$\hat{H}^{CT-vib} = \sum_{k=1}^{N_{CT}} \sum_{\alpha}^{N_{cat}} \lambda_k^{\alpha, cat} q_{\alpha} |CT_k\rangle \langle CT_k| + \sum_{k=1}^{N_{CT}} \sum_{\alpha}^{N_{ani}} \lambda_k^{\alpha, ani} q_{\alpha} |CT_k\rangle \langle CT_k| \quad (\text{S21.6})$$

$\lambda_k^{\alpha, (LE, ani, cat)}$ represent the gradients of the excited potential energy surface (excitonic, cationic or anionic states of the CT) at the equilibrium geometry of the ground state and are referred to as first-order intra-state electronic-vibrational coupling constants (which related also to the dimensionless shift g_k^{α} by $\lambda_k^{\alpha} = g_k^{\alpha} \hbar \omega_{\alpha}$).³⁶ In Eq. S21.6, we make the assumption that the CT

states are simply given by the formation of cationic state on the DPP-BDT polymer with and the anion on the PDI unit.

Diabatization and Electronic Hamiltonian

To carry out the quantum dynamics simulations, we first constructed the electronic Hamiltonian in Eq. S21.4 in the diabatic representation. This was achieved performing electronic structure excited state calculations on a reduced model comprising a single polymer (DPP-BDT) unit and a single PDI molecule. The use of this simplified model was motivated by computational efficiency and by the fact that the adiabatic states of this reduced system (supplementary fig. S48) are representative of those in larger polymers (supplementary fig. 36). This is also shown in supplementary table 3, with the most important difference being the number of states with CT character in the same energy window (which is higher for longer polymer systems).

Supplementary Fig. 48 | Excitation energies, oscillator strengths, and dipole moments of the first three excited states of the optimized TS-P3 monomer structure, along with the corresponding hole-particle natural transition orbitals.

Supplementary Table 3 | Excitation energies (in eV) and their associated oscillator strengths for the TS-P3 dimer and monomer, as computed at the TDDFT SRSH LC- ω hPBE/6-311G(d,p) level of theory.

TS-P3 dimer			TS-P3 monomer		
State (character)	Energy (eV)	Osc. Str.	State (character)	Energy (eV)	Osc. Str.
S1 (CT1)	1.23	0.009	S1 (CT1)	1.36	0.018
S2 (CT2)	1.33	0.025	S2 (CT2)	1.96	0.002
S3 (CT3)	1.68	0.020	S3 (FE)	2.15	0.376
S4 (CT4)	1.81	0.009			
S5 (CT5)	1.87	0.003			
S6 (FE)	1.94	1.776			

As shown in Fig. S48 and Table S3, our TS-P3 monomer model, optimized at the wB97X-D/6-31G(d,p) level of theory, features one bright excitation localized on the polymer and two lower-energy dark states corresponding to charge-transfer (CT) states from the polymer donor to the PDI acceptor. To evaluate the electronic couplings between the diabatic LE and intramolecular CT states, we employed an adiabatic-to-diabatic transformation using the multi-state fragment excitation difference–fragment charge difference (MS-FED-FCD) method. The MS-FED-FCD method extends and unifies the capabilities of previously developed property-based diabaticization schemes, namely the two-state fragment excitation difference (FED)³⁷ and fragment charge difference (FCD)^{38,39} approaches, which have been described in detail elsewhere^{40–43}. The method partitions the system into two fragments, a donor and an acceptor (in our case the DPP-BDT unit and the PDI, respectively), and, using suitable additional operators, transforms the adiabatic Hamiltonian of the dimer (comprising two or more adiabatic states) into a diabatic representation yielding the electronic Hamiltonian in Eq. S21.2. This transformation enables direct evaluation of the excitonic couplings between LE and CT states from the diabatic Hamiltonian matrix.

Here, we applied the MS-FED-FCD diabaticization in combination with TDDFT at the LC- ω hPBE/6-311G(d,p) level of theory to both the monomer model TS-P3 system and the dimer TS-PX reported in the main text and supplementary note 15. Our simulations (see supplementary table 4) confirm that the electronic couplings between LE and CT states are small (i.e., 15-28 meV) in both cases, consistent with our experimental observations of weak excitonic effects influencing the optical absorption. Therefore, the ultrafast charge separation observed is not driven by strong excitonic couplings as it is the case for other system in the literature^{44–46}. As discussed in the main text and shown by full quantum dynamics below, even in the presence of such weak electronic interactions, rapid charge separation occurs, driven by vibronic couplings and energy alignment between LE and CT states.

Supplementary Table 4 | Diabatic excitation energies (in eV) for the TS-P3 dimer and monomer, as computed with the multi-state fragment excitation difference–fragment charge difference (MS-FED-FCD) method on top of a TDDFT SRSH LC- ω hPBE/6-311G(d,p) calculation. In the table there are also reported the electronic coupling (in meV) between the Frenkel exciton (FE) and the various CT states.

TS-P3 dimer	TS-P3 monomer
-------------	---------------

	Energy (eV)	FE-CT coupling (meV)		Energy (eV)	FE-CT coupling (meV)
CT-1	1.23	15	CT-1	1.35	15
CT-2	1.33	-25	CT-2	1.96	28
CT-3	1.68	10	FE	2.20	/
CT-4	1.81	-1			
CT-5	1.87	3			
FE	1.96	/			

Supplementary Table 5: Diabatic Hamiltonian of the TP-P3 monomer with and without linkers. All the elements are in eV.

	TS-P3 mon.			TS-P3 mon. without linkers		
	LE	CT2	CT1	FE	CT2	CT1
FE	2.196	0.015	0.028	2.211	0.015	0.027
CT2	0.015	1.930	0.000	0.015	1.986	0.000
CT1	0.028	0.000	1.346	0.027	0.000	1.355

Our diabatic Hamiltonian shows that, as we move from low-energy charge-transfer (CT) states (e.g., CT1) to higher-energy CT states, the excitonic coupling decreases in both monomer and dimer polymers. This behaviour arises because the electron and hole become more spatially separated, as illustrated in supplementary fig. 48 and 36. The larger separation leads to a weaker Coulomb attraction, which in turn results in a higher energy for the long-range CT states. We also note that the energy offset between FE and CT states in our simulations depends on the polymer size. Specifically, the energy offset decreases as the polymer length increase as we show in Table S4.

Supplementary Figure 49 / Representation of TS-P3 monomer with and without non-conjugated linkers connecting the DPP-BDT unit and the PDI

As a further note, to understand whether the linkers connecting the DPP-BDT unit to the PDI affect the excitonic properties of the aggregate, we also performed the diabatization in the same TS-P3 model system where the non-conjugated linkers between donor and acceptor were removed (Supplementary Fig. 49). The corresponding Hamiltonians are reported in supplementary table S5. Both cases yield similar electronic couplings, confirming that the non-conjugated linkers play a negligible role and that the interactions are primarily through-space in nature.

Normal mode analysis and Vibronic interactions

As observed in our experiments, the vibronic interactions play an important role in the ultrafast dynamics of the TS-P3 systems. To estimate the strength of these interactions, we computed the gradients λ_k^α along the normal mode coordinates appearing in Eq. S21.5 and S21.6 using the normal modes of the isolated structures if DPP-BDT and PDI fragments taken at the optimized TS-P3 monomer geometry in its ground state in the gas phase (see Fig. S48). These gradients, used to set-up the LVC Hamiltonian in Eq. S21.4, are computed analytically employing a (quantum) harmonic oscillator model. This approach allows us to compute the

ground state potential energy surface (PES) in the harmonic approximation using the QM hessian and frequency at the isolated fragment geometries.

To build the excited-state PES, which in our case, refer to the locally excited state and the cationic state of the DPP-BDT molecule and the anionic state the PDI molecule, respectively, we adopt the Vertical Gradient (VG) approach as implemented in the FCclasses3.0 software.^{47,48} Within this approximation final-state PES is assumed to have the same normal modes and frequencies as the initial (ground) state potential. The VG accounts only for the effect of the dimensionless displacement g_α , for each mode α from the equilibrium position. These displacements are related to the gradients λ_k^α by: $\lambda_k^{\alpha,(\text{LE,ani,cat})} = g_k^\alpha \hbar \omega_\alpha$ for a given k state. The frequencies are multiplied by a factor 0.949 as done for the TS-P3 dimer in supplementary fig. 36.

The gradients calculated in this way for DPP-BDT LE and cationic states as well as for the anionic PDI states are reported in Fig. S50. The relaxation energy of each mode can be estimated as well $\lambda_\alpha^{rel} = \frac{g_k^{\alpha^2}}{2} \hbar \omega_\alpha$ and it is reported in the same Figure.

We note that, in general, the gradients of different CT states are not identical. However, for simplicity, we assume here that all CT states have the same gradients, obtained by summing the gradients of the anionic and cationic contributions as shown in Eq. 21.6. This approximation could be relaxed by computing the gradient numerically for each CT state.^{35,49} However, this is not practical, as it would require diabating the states for each normal mode of the TS-P3 monomer, which contains a large number of modes. For this reason, we rely on the simpler approach described above.

These simulations clearly show that several high-frequency modes of the DPP-BDT are strongly coupled with both the exciton as well as the cationic states. In particular, the mode at 1531 cm^{-1} (at a frequency that agrees with the mode observed in IVS, 1519 cm^{-1}) shows the largest displacement and can be considered the driving mode of this reduced model. Concerning the PDI, our simulations show a strong coupling between the anion and the modes around $\sim 1200\text{-}1300 \text{ cm}^{-1}$. This observation support the fact that the vibrational coherence along the mode at 1283 cm^{-1} is generated upon the electron transfer. As we discuss below, these modes are clearly involved in the excited state dynamics of the system which support the experimental observed in the impulsive vibrational spectroscopy.

Supplementary Fig. 50 | Representation of the structures of a) DPP-BDT unit and b) PDI unit used to compute the gradients (middle panels) of the PES of exciton and cation of DPP-BDT and anion of PDI, respectively as well as related relaxation energies (bottom panels) as explained in the text.

Quantum dynamics

Nuclear wave-packet propagations were carried out using the MCTDH method²⁹⁻³³, as implemented in the QUANTICS code.⁵⁰ The wavepackets were propagated for 200 fs in steps of 0.5 fs. As the number of modes is sizable, we relied on multi-Layer (ML) extension of MCTDH.³³ We adopted the variable mean field scheme with a Runge-Kutta integrator of order 5 and an accuracy threshold of 10^{-7} . Different numbers of single-particle functions (SPFs) for each layer as well as a different number of modes were tested for convergence of the results.

Notably, considering all normal modes for each diabatic state of TS-P3 monomer is still too demanding. To reduce the number of modes included in the Hamiltonian (Eq. S21.1), we selected a total of 36 most important modes (18 for the DPP-BDT and 18 for the PDI), with the largest contribution to the total relaxation energy. The selected modes and their corresponding gradients are reported in Table 6.

Table 6: Electron vibrational couplings and related relaxation energies of the 36 modes included in the LVC Hamiltonian for DPP-BDT excitonic and cation state as well as the PDI anionic state

DPP-BDT					PDI		
Exciton			cation		anion		
ω_α (cm ⁻¹)	λ^α (eV)	λ_α^{rel} (eV)	λ^α (eV)	λ_α^{rel} (eV)	ω_α (cm ⁻¹)	λ^α (eV)	λ_α^{rel} (eV)
1531.1	0.1543	0.0627	0.1197	0.0378	1598.8	0.1388	0.0486
1361.5	0.1050	0.0327	-0.1092	0.0353	1373.0	-0.1077	0.0341
1726.0	0.0214	0.0011	0.0876	0.0179	1353.5	0.0882	0.0232
14.0	0.0045	0.0059	0.0078	0.0173	387.5	-0.0342	0.0122
179.5	-0.0023	0.0001	-0.0240	0.0129	1579.4	0.0595	0.0090
472.3	0.0222	0.0042	-0.0347	0.0103	1753.0	0.0606	0.0084
175.6	0.0074	0.0013	0.0210	0.0101	1255.2	0.0500	0.0080
1066.2	0.0496	0.0093	0.0502	0.0095	1044.2	0.0451	0.0079
64.7	-0.0013	0.0001	-0.0119	0.0089	526.5	0.0292	0.0065
40.5	0.0090	0.0081	-0.0081	0.0065	1298.1	-0.0439	0.0060
607.1	-0.0170	0.0019	-0.0347	0.0080	1743.0	0.0477	0.0053
1224.2	0.0125	0.0005	0.0467	0.0072	1296.7	0.0407	0.0052
1438.9	-0.0401	0.0045	0.0487	0.0066	1431.3	0.0361	0.0037
619.1	-0.0134	0.0012	-0.0309	0.0062	226.9	0.0136	0.0033
690.4	0.0261	0.0040	0.0317	0.0059	802.9	0.0138	0.0010
1496.2	0.0464	0.0058	-0.0240	0.0015	707.0	0.0120	0.0008
1443.2	0.0323	0.0029	-0.0417	0.0049	22.6	-0.0021	0.0008
1109.5	-0.0159	0.0009	-0.0350	0.0045	1736.9	-0.0157	0.0006

The results of our quantum dynamics (QD) trajectories are reported in Fig. 51. The first important observation is that a fast population transfer is recovered when using the Hamiltonian defined in supplementary table 5 with an energy offset between the FE state and the CT1 and CT2 of 0.850 eV and 0.583, respectively as found for our model TS-P3 monomer system (see yellow line in supplementary fig. 51). However, this transfer remains slower than that observed experimentally and an important reason for that is that as discussed and shown before in Table S4, longer polymers show a reduced offset between FE and CT states. Thus, we tested the sensitivity of the electron transfer with respect to such an offset, by decreasing the energy difference between LE and CT2(1) linearly by factor 2. We find that the rate of this ultrafast transfer is strongly dependent on such an energy offset between the LE and CT states. The transfer gets faster when it becomes barrierless.

We note that when using an offset value comparable to the experimental estimate (around 200 meV), the transfer rate decreases to ~ 75 fs, thus closing the gap with the experimental measurements. Remaining discrepancies might be attributed to the approximation of using the

same gradients for different CT states, the reduced number of nuclear degrees of freedom and the absence of non-local electron-vibrational couplings.

Despite these differences, our simulations clearly show the rapid generation of CT states that get populated on an ultrafast time scale. More importantly, they reveal that the fast-driving mode of the DPP-BDT polymer at 1531 cm^{-1} , together with contribution from the other vibrational modes, drives the initial ultrafast transfer. The PDI modes, in contrast, begin to evolve only after the CT states are populated via this fast electron transfer, in agreement with the experimental observations. This is especially clear for the PDI mode at 1299 cm^{-1} , whose frequency coincides with what has been found experimentally. In our impulsive vibrational spectroscopy measurements, the vibrational coherence along the 1283 cm^{-1} coordinate appears only after electron transfer, fully consistent with the simulations, which identify this mode as a PDI-anion vibration activated following CT-state formation.

Supplementary Fig. 51: a) Population decay of the FE state and its fitting with a monoexponential function to extract the characteristic time scale of population transfer for different energy offsets between the L and CT2(1) states. b) Excited-state population for an energy offset of approximately 200 meV between the FE and CT states, as observed experimentally. c) Average wavepacket position as a function of time. This shows that the high-frequency DPP-BDT mode begins oscillating immediately with a period of ~ 20 fs, whereas the PDI modes (particularly the one at 1299 cm^{-1}) start oscillating only after the population transfer has occurred.

Other minor comments:

There seems to be significant contribution of cross-phase modulations close to time zero especially seen from crosscuts extending over 100 fs. Therefore, it becomes important to show the subtraction scheme especially for analysing data in 20-40 fs waiting time.

Response: We thank the reviewer for pointing this out. We have addressed this point while addressing the question number 2.

Action: We have included the subtraction method of the kinetic to exclude the cross-phase modulation before kinetic fitting which is included in the supplementary information note 17.

Page 10, line 238, the reference to figure, should read Figure 3b?

Response: We thank the reviewer for noticing this error.

Action: The reference has been corrected to “figure 3b” from figure 4b in the revised manuscript.

Page 3, line 63, factors that govern

Response: We thank the reviewer for pointing this mistake out.

Action: The text has been corrected from “factors that governs” to “factors that govern” in the revised manuscript.

Reviewer #2 (Remarks to the Author):

The manuscript by Ghosh, et al. details the excited-state dynamics of model polymer/non-fullerene acceptor heterojunctions monitored with high-time resolution broadband transient absorption spectroscopy. The model systems are comprised of a dyketopyrrolopyrrole-benzodithiophene (DPP-DBT) polymer donor and a perylene diimide (PDI) acceptor in two different configurations: one with the PDI closely interacting with the electron-rich BDT unit (TS-P2), and the other where it is more closely associated with the electron-deficient BDP moiety (TS-P3). They observe sub-picosecond charge separation in TS-P3, while for TS-P2 they show charge separation in <15 fs, which is less than the vibrational period of the high-frequency Raman-active transitions available. The results are high quality and the analysis is sophisticated, and the understanding proposed will be of significant interest to the field. While the proposed explanation is

compelling, there are numerous unanswered questions that need to be addressed before this work can be considered publishable in Nature Communications.

Response: We thank the reviewer for their thoughtful and constructive evaluation of our manuscript. We are especially encouraged by the positive remarks that “*The results are high quality and the analysis is sophisticated, and the understanding proposed will be of significant interest to the field.*” We appreciate that the reviewer recognises the novelty of our model donor–acceptor systems and the mechanistic insights into ultrafast charge separation that we provide. Their acknowledgement of the sophistication of our analysis and the potential impact of our findings on the wider community is highly motivating.

Major issues:

To address the queries point by point we have divided the first comment into multiple sub-comments.

-The authors central claim is that charge transfer here is fast while still directly occurring from the local exciton state of the polymer and not an optically accessible charge-transfer state. They have performed photothermal deflectance spectroscopy measurements and DFT calculations to rule out this possibility.

We agree with the reviewer that one of the key output of this work is formation of the CT state in ultrafast sub-vibrational period timescale directly from the local exciton state of the polymer without accessing a optically accessible charge transfer state. The key supporting evidence for the information is as reviewer mentioned

- Absence of the sub-bandgap CT transition revealed by the photothermal deflection spectroscopy (PDS)
- By exciting the sample with broadband red pump when the CT transitions are in turn $\sim 10^2$ weaker than the singlet LE transition.
- Additionally we don't observe any PDI-specific mode during the cw-Resonant Raman spectroscopy with 785 nm excitation. (figure 3b(5))
- We also don't observed any PDI-specific mode at the PDI ground state bleach regime (fig. 3b(6)). We only observed polymer backbone modes.

Action: We have included these four key observations, which support the conclusion that direct excitation of a charge-transfer (CT) state does not occur, in the decision summary table in the supplementary information 44 which is added in the revised manuscript.

However, the photothermal deflectance spectroscopy was performed under quite different conditions (i.e. in a high polarity environment and in a solvent where aggregation is discouraged), and so this is not conclusive.

Response: We thank the reviewer for raising this important point and agree that solution-phase PDS differs from the more commonly implemented film-based measurements, particularly with respect to aggregation and local morphology. In our study, we intentionally performed PDS in chloroform solution—following the methodology of Sun *et al.* (*Nature* **615**, 830–835, 2023)—to ensure full consistency with our transient absorption and impulsive vibrational experiments, which were all carried out in the same solvent environment.

We acknowledge that film-based PDS could provide complementary information; however, it would also introduce additional inter-chain aggregation- and morphology-dependent variables that could obscure the specific donor–acceptor orientation effects we aimed to isolate. For this reason, we selected the solution-phase configuration to provide a cleaner, internally consistent comparison between the other experiments such as IVS, TA performed on the two model heterojunction studied here.

Action:

The following part has been added in the method section of the main manuscript.

Revised main manuscript excerpt

“Photothermal deflection spectroscopy (PDS) was performed in chloroform solution, following the approach of Sun *et al.* (*Nature* 615, 830–835, 2023). We chose the solution-phase method to ensure consistency with the solvent environment used in our transient absorption and vibrational experiments, and to avoid additional aggregation or morphology effects that can complicate interpretation in thin-film measurements.”

The DFT calculations are also prone to functional- and basis set-specific artifacts that could miss such interactions.

Response: We agree with the reviewer that DFT has its own drawbacks, especially when used for excited-state calculations, and might miss some important effects depending on the choice of the exchange-correlation functional. However, in this work, we were interested in a qualitatively accurate description of the CT and Frenkel excitonic states, and for that reason, we employed a state-of-the-art scheme, which uses a range-separated hybrid functional (specifically designed for the characterization of long-range effects as CT states, which are not fully captured by hybrid functional like B3LYP) and introduces screening effects to mimic the environment. We are thus confident that the interactions the reviewer is referring to are well included in our method. Regarding the basis set, for excited-state calculations we used a 6-311G(d,p) basis set, which is a quite large basis set, even considering the size of our system (> 250 atoms). Our data suggests that oscillator strengths of the charge-transfer (CT) transitions are roughly 800 times weaker than those of the lowest Frenkel excitonic transitions, supporting the assignment of the 1283 cm^{-1} vibrational coherence as not originating from any directly excitable CT transition, as shown below.

Action:

Supplementary Fig. 2.2 Calculated electronic transitions of the DPP-BDT dimer of the TS-P3 architecture. Asterisked mode corresponds to the Frenkel exciton (FE). excited-state calculations is performed using 6-311G(d,p) basis set with range-separated hybrid functional (specifically designed for the characterization of long-range effects as CT states) which further reveal oscillator strengths of the charge-transfer (CT) transitions are roughly 800 times weaker than those of the lowest asterisked Frenkel excitonic transitions.

Generally, the UV/vis spectra usually give some indication of donor-acceptor interactions that give rise to new CT bands, but these data (Figure 1c) are not discussed in a

meaningful way in this context. Specifically, there are substantial differences between the absorption spectra of TS-P2 and the constituent Ref-P/PDI spectra that are not discussed. The splitting and red-shifting of the low-energy polymer absorption band could easily indicate aggregation of the polymer backbone or charge-transfer interactions between the PDI and BDT. Similarly, though less drastically, the absorption of TS-P3 shows broadening of the vibronic progressions of both PDI and the polymer transitions. These differences in spectra lead to questions about aggregation and disorder that should be addressed: how rigid are these structures and what role does flexibility and overall disorder have on the dynamics? Are the polymer molecular weights (including those within the distribution) sufficient to allow additional interactions with the PDI unit? A discussion of the length of the polymer in the main text would add to the rigor of the discussion of the absorption spectrum. But moreover a better discussion of the UV/vis spectra and whether the clear differences in the spectra are indicative of ground state electronic interactions should be given.

Response: For TS-P2 and TS-P3, the spectrum can be divided into the 450–570 nm (2.2–2.7 eV) region, where the PDI chromophore absorbs, and the 570–800 nm (1.5–2.2 eV) region, which corresponds with absorption from the conjugated polymer backbone. Both Ref-P2 and TS-P3 have absorption maxima at ~760 nm (~1.63 eV) with a vibronic shoulder at ~690 nm (~1.80 eV). The absorption spectrum of TS-P2 shows an inversion of the “0–0” and “0–1” intensities, such that the maximum is found at ~690 nm with a smaller peak at ~760 nm. We have previously studied the origin of the inversion of these absorption peaks where we reported the initial synthesis paper¹. As shown in Fig 5 and 6 of the supplementary information in this manuscript¹, the different ratios of these two peaks was captured using our xtb-ensemble approach which included vibronic contributions. Importantly, the difference in ratios of the two absorption bands was dictated by the conformational changes to the polymer backbone and not any intermolecular (or alternatively same polymer chain intramolecular) aggregation effects. Indeed we also found that TS-P3 showed greater conformational disorder than TS-P2 (Fig 7 – SI of that paper¹). In addition to this, we also reported the concentration dependent UV-Vis and we did not observe an substantial changes further suggesting that this is not an intermolecular effect. Finally, we note that when planning the synthesis of these

materials, we introduced the bulky chain on the rear of the PDI to discourage any intramolecular PDI stacking. The reviewer asks an interesting question with respect to the role of flexibility and disorder on the dynamics. Currently this work is ongoing and subsequent generations of these materials are being designed to possess similar energetics + distance but different levels of rigidity but these materials are extremely complex to synthesize and beyond the scope of this investigation.

Action:

We have added a sentence in the main text saying

“The origin of the different intensities of the 0-0 and 0-1 peak in the UV-Vis absorption spectra have been discussed previously (reference¹) and we believe they predominantly arise for a greater degree of conformation disorder in TS-P3 relative to TS-P2.”

-It is not clear how these results translate into solid-state design rules for better organic photovoltaic (OPV) device performance. Rapid charge separation from polymers to non-fullerene acceptors, including PDI is not novel and not inherently reliant on the mechanism proposed here. While this mechanistic approach will certainly be interesting to the community, it is not clear how these effects can be leveraged in devices. Better contextualization of these results is necessary to increase their impact and utilization by the community. Related, there are significant examples in Nature where fast, high yielding charge transfer occurs with minimal energy loss, such as the special pair in photosynthesis. This strategy has even been adapted in artificial systems for OPVs utilizing symmetry-breaking charge transfer [J. Am. Chem. Soc. 2015, 137, 16, 5397–5405]. A difference, of course, is that in those systems there are frequently indications of strong electronic coupling in the ground state, except in the case of null-aggregates which are known to exhibit excited-state symmetry breaking. In those and related cases, the role of vibrations has been investigated [J. Am. Chem. Soc. 2022, 144, 34, 15539–15548, Nat. Chem. 14, 786–793 (2022)], but this context is lacking here. As such, the notion that energy loss and charge transfer rates are "fundamental barriers" are overstatements.

Response: We thank the reviewer for this insightful and constructive comment, which has greatly strengthened the conceptual framing and broader impact of our manuscript. We fully agree that positioning our results alongside the well-established framework of *symmetry-breaking charge transfer* (SBCT) and related vibronic-coupling phenomena provides valuable context. Seminal contributions from Kim, Wasielewski, Würthner, Thompson and co-workers (*J. Am. Chem. Soc.* 2015, 137, 5397; *Nat. Chem.* 2022, 14, 786; *J. Am. Chem. Soc.* 2022, 144, 15539) have elegantly shown that strong ground-state electronic coupling, and in some cases null-aggregate configurations, can enable the generation of charge-transfer states with minimal energy loss. In contrast, our study extends this mechanistic concept to *weakly coupled polymer–acceptor heterojunctions*, which lack measurable ground-state coupling or aggregate order. The model system is deliberately designed using a low-bandgap push–pull polymer and a non-fullerene acceptor with minimal energetic offset, yielding regio-selective and exceptionally rapid (< 15 fs) electron transfer. Here, the transfer proceeds through coherent mixing of the locally excited (LE) and charge-transfer (CT) states in the excited manifold, rather than through ground-state delocalisation. In TS-P3, selective coupling between the donor’s high-frequency (1512 cm⁻¹) DPP-localised vibration and the electronic gap drives this sub-vibrational-period transfer, demonstrating that coherence-mediated charge transfer can occur even in electronically disordered systems. We believe this observation complements and broadens the SBCT paradigm—showing that *mode-specific vibronic coupling* can be purposefully engineered to promote ultrafast, low-loss charge separation in realistic solid-state environments where strong static coupling is absent. In response to the reviewer’s valuable feedback, we have revised both the *Introduction* and *Discussion of the results* to explicitly link our findings to the SBCT framework, highlight their relevance for OPV material design, and acknowledge the pioneering work of the above-cited studies. We have also refined the phrasing from “fundamental barriers” to “prevailing limitations” to maintain a balanced perspective. We are grateful to the reviewer for prompting these additions, which have notably improved the clarity, rigour, and contextual significance of the paper.

Action

- The Introduction has been revised to clearly contextualise the symmetry-breaking charge transfer (SBCT) mechanism in relation to efficient charge-transfer (CT) state formation with minimal energy loss.
- The Discussion now explicitly connects the present findings to the broader SBCT and vibronic-coupling literature, with appropriate citations to key studies.
- The phrase “fundamental barriers” has been rephrased as “prevailing limitations” to maintain a balanced and precise interpretation.

Revised main manuscript excerpt

Introduction section: Traditionally, the rate of charge transfer in light harvesting devices is believed to be primarily governed by the electronic coupling and energy offset between the frontier orbitals between donor and acceptor molecules^{51,52}. *Recent studies on symmetry-breaking CT (SBCT) in covalent dimers and ordered aggregates have shown that strong electronic coupling can enable ultrafast, low-energy-loss CT^{44–46}*. However, the role of vibrational modes—whether they assist, limit, or modulate the charge transfer process in weakly coupled organic heterojunctions—remains poorly understood. Understanding this interplay is essential for identifying the fundamental speed limits of photoinduced charge separation in molecular heterojunction based photovoltaic devices.

Discussion section: In contrast, the FE energy plunges when moving along the 1529 cm⁻¹ mode leading to (avoided) crossings with the lower-lying CT excitations and a massive reorganization of the oscillator strength, see Figure 4b. In a way, this is expected since this vibrational mode has a large amplitude over the regions of the backbone in close contact with the PDI acceptors. Thus, the theoretical calculations support the view that the 1529 cm⁻¹ mode drives the electron transfer from the polymer backbone to the nearby PDI units, by causing a mixing of the singlet exciton and CT PES surfaces. This is illustrated in Figure 4c. Hence it is this driving mode that undergoes a large reduction of intensity in the IVS spectra. *This mechanism is conceptually analogous to the vibronic coherence driven coupling between locally excited and charge-transfer states reported in symmetric molecular dimers showing symmetry-breaking charge transfer (SBCT)^{45,46},*

but here it operates within an asymmetric, weakly coupled heterojunction which lacks measurable ground state coupling.

Conclusion section:where our results suggest that previously established design rules about the tradeoff between energy loss and charge transfer rates may not represent fundamental barrier-prevaling limitations to improve further materials design strategies.

-The ultrafast nature of the charge transfer in TS-P3 being driven by the 1512 cm⁻¹ polymer mode is intriguing, so the natural question is how general is this conclusion? The authors state that this mode is localized mostly in the DPP unit, which seems consistent with the difference in dynamics between TS-P2 and TS-P3, however, this is not emphasized in the discussion. Indeed, it This is disappointing, as it may point towards the generality of this mechanism and its use in designing more efficient charge transfer systems. This element should be discussed more. Does the solvent play any specific role, either through inertial or dielectric stabilization or through coupling to the solute modes? Such effects would not be present in the solid-state device so this potential effect is worth clarification.

Response:

We thank the reviewer for this thoughtful and constructive comment. We agree that assessing the generality and environmental dependence of the 1512 cm⁻¹-driven charge-transfer (CT) mechanism is crucial for evaluating its broader relevance to organic photovoltaic (OPV) systems.

We would like to emphasize that our present study is aimed at understanding the *fundamental interfacial processes* that determine the initial charge-separation dynamics, rather than optimizing device-level power conversion efficiency. Modern OPV heterojunctions typically consist of low-bandgap push-pull polymers paired with non-fullerene acceptors, giving rise to multiple interfacial configurations. This structural heterogeneity is deliberately mimicked in our two model heterojunctions, TS-P2 and TS-P3. As the reviewer correctly notes, the distinct behaviour of these two systems demonstrates the robustness of our conclusions: in TS-P3, where the PDI acceptor is positioned adjacent to the electron-deficient DPP unit, the DPP-localised 1512 cm⁻¹

mode couples directly to the electronic gap, driving sub-15 fs electron transfer. In contrast, in TS-P2, where the DPP moiety is spatially separated from the PDI, this vibronic coupling is much weaker, resulting in slower (≈ 376 fs) charge transfer. Thus, the *generality* of the mechanism lies in this regio-specific control — when the driving vibrational mode is co-localised with the donor–acceptor interface, ultrafast charge transfer is universally enabled. As the reviewer suggested this element is discussed in the revised manuscript.

Regarding the influence of the solvent, all measurements were performed in a non-coordinating, weakly polar medium. Exploring solvent- and linker-dependent effects is an active direction of our ongoing work, involving variations in both the solvent polarity and the length of the non-conjugated tether between donor and acceptor. These results will be presented in a forthcoming publication. However, it is worth emphasizing that this study focuses on the *fundamental role of interfacial arrangement* in governing charge-transfer dynamics using well-defined model heterojunctions. While these systems are designed to emulate the key electronic configurations found in real OPV interfaces, replicating such precisely controlled molecular geometries in device-scale architectures lies beyond the scope of the present work. This clarification has been added to the revised manuscript.

Action:

Revised main manuscript excerpt

Introduction section:which we will refer to as TS-P2 (‘acceptor-on-electron rich system’) and TS-P3(‘acceptor-on-electron deficient system’) architectures, respectively (Figure 1b, for synthetic protocols see supplementary methods and ref¹). These model systems allow us to isolate the dynamics of the CT process, which can be masked via exciton and charge-transport processes in OPV thin-films or by molecular diffusion in photocatalytic systems^{53–56}. **We emphasize that this study aims at elucidating the fundamental interfacial processes governing initial charge-separation dynamics, rather than optimising device-level power-conversion efficiency.**

Discussion section: The regio-specific contrast between TS-P3 and TS-P2 underscores the generality of this vibronic-driving mechanism. When the high-frequency DPP-localised vibration ($1512\text{--}1529\text{ cm}^{-1}$) coincides with the donor–acceptor interface, as in TS-P3, it directly modulates the CT gap, enabling sub-15 fs electron transfer. In TS-P2, where this mode is spatially displaced from the PDI acceptor, the coupling is weaker and CT slows markedly ($\approx 376\text{ fs}$). This demonstrates a general design rule: co-localising interfacially active vibrational modes with the electronic coupling region enables ultrafast CT even in weakly coupled systems.

-The CW Raman spectra results for PDI where the 1283 cm^{-1} mode is absent are compelling evidence for the lack of ground-state excitation of the PDI. However, the signal-to-noise ratio of these data is not particularly high and there may be indications of this missing feature close to the noise floor.

Response: We thank the reviewer for this helpful comment and for recognising that the absence of the 1283 cm^{-1} mode in the cw-Raman spectrum supports the lack of direct ground-state excitation of any charge-transfer complex involving the PDI moiety. We have repeated the 785 nm excitation cw-Raman measurements, and the unsmoothed raw data are now included in Supplementary Fig. 45. We agree that the signal-to-noise ratio is inherently lower than for 532 nm excitation, as expected from the $1/\lambda^4$ dependence of Raman scattering efficiency ($\approx 5\times$ weaker at 785 nm) and the presence of a fluorescence background. Nevertheless, the new measurements—performed under multiple experimental configurations—consistently show no detectable 1283 cm^{-1} feature within the experimental noise limits, reinforcing our original conclusion. This finding is further supported by the 532 nm cw-Raman data, where strong PDI modes are clearly observed even at 5% PDI loading suggesting a strong Raman cross-section of that mode when it is resonantly excited. This supports that any ground-state CT feature at 785 nm would be well above the current detection limit if it existed. To aid clarity, we have also included a decision-tree summary in the Supplementary Information (supplementary Fig. 44) that

outlines all experimental and analytical steps used to assign the 1283 cm⁻¹ mode to the excited-state PDI radical anion.

Action:

Revised main manuscript excerpt

Utilizing continuous wave resonance Raman with 785 nm excitation on TS-P3, we do not observe the ω^β (1283 cm⁻¹) mode (Figure 3b(5)) , even though the PDI modes—including the analogous 1291 cm⁻¹ band—exhibit strong Raman cross-sections when excited resonantly (Supplementary Note 19).

Revised supplementary information excerpt

Supplementary Note 19: Additional cw-Raman Analysis of PDI and TS-P3 Derivatives

To verify whether the 1283 cm⁻¹ mode observed in the time-resolved measurements could originate from directly excitable PDI species (any *ground-state charge-transfer (CT) complex*), we performed comparative continuous-wave (cw) Raman spectroscopy using both 532 nm (resonant with PDI) and 785 nm excitation which should be resonant o any ground state CT complex if exists.

The spectra of TS-P3, TS-P3-5 %, and TS-P3-10 % (where PDI loading is reduced to 5 % and 10 %, respectively, please see Supplementary Fig. 7 for chemical structure) exhibit pronounced resonance-enhanced bands associated with the PDI chromophore (marked by red asterisks). Even at 5 % loading, the PDI modes strongly dominate over the polymer backbone features (black asterisks), confirming the strong Raman cross-section of PDI modes under resonant excitation. The strong enhancement of these bands at very low concentrations demonstrates the high sensitivity (Raman cross section) of the cw-Raman experiment to PDI vibrations when resonantly excited.

Under 785 nm excitation, TS-P3 spectra show markedly weaker overall Raman intensities, as expected for off-resonant conditions where the Raman scattering efficiency scales as $1 / \lambda^4$ ($\approx 5\times$ weaker at 785 nm than 532 nm). The signal-to-noise ratio is further reduced because of the fluorescence background. Importantly, no Raman feature is detected near 1283 cm⁻¹ within the 3σ noise level, even in the TS-P3 spectrum

(black trace, Fig. 45). The noise level was estimated from the flat baseline region (1700–2000 cm^{-1}), confirming no significant band at 1283 cm^{-1} with an amplitude $> 3\sigma$.

The combination of (i) strong, clearly detectable PDI modes under resonant excitation even at 5 % PDI content and (ii) the complete absence of any detectable feature near 1283 cm^{-1} at 785 nm off-resonant excitation provides compelling evidence that no ground-state CT complex or directly excitable PDI species exists with an appreciable cross-section near 785 nm. If such a complex were present, the 1283 cm^{-1} mode would be expected to under these conditions.

These results therefore confirm that the 1283 cm^{-1} vibrational coherence observed in the transient absorption experiments originates not direct excitation of the PDI involved CT complex.

Figure 45. cw-Raman spectra of PDI and TS-P3 under 532 nm and 785 nm excitation. Resonant 532 nm excitation (blue/orange) shows strong PDI modes (red asterisks), even at 5 % loading, while off-resonant 785 nm excitation (black) shows no detectable 1283 cm^{-1} feature within the 3σ noise level (estimated from 1700–2000 cm^{-1}). This confirms the absence of any directly excitable ground-state CT complex.

-Can the authors perform a short-time Fourier transform on the transient absorption data to see how the vibrational modes behave in time [J. Am. Chem. Soc. 2022, 144, 15539–15548]? This may better help demonstrate their proposed mechanism.

Response: We thank the reviewer for this valuable and constructive suggestion. We performed a time-domain analysis of the vibrational modes after applying narrow-band bandpass filtering. The results reinforce our earlier finding that the 1512 cm⁻¹ mode—previously identified as the driving vibration for electron transfer from CW-resonant Raman, resonant IVS, and quantum-chemical analyses—exhibits a markedly faster dephasing time. Following the reviewer’s recommendation, we also conducted a short-time Fourier transform (STFT) analysis using an approach consistent with the cited reference. The TS-P3 IVS data reveal a clear lag in the appearance of the 1283 cm⁻¹ mode, supporting its assignment as a vibration generated by the electron-transfer process rather than by direct impulsive photoexcitation. Owing to space constraints, a detailed description of these analyses is provided in a dedicated section of the Supplementary Information, which is now cited in the main manuscript.

Action:

Revised main manuscript excerpt

“The short time FT analysis (Supplementary Fig. 47) shows a clear temporal lag in the appearance of the ω^β mode, confirming that it arises from charge-transfer dynamics rather than direct impulsive excitation. ”

“This is further supported by the faster decoherence of the 1507 cm⁻¹ mode in TS-P3 relative to Ref-P, whereas the other Franck–Condon-active modes exhibit comparable dephasing times in both systems (supplementary figure 46) in inverse FT analysis.”

Revised supplementary information excerpt

Supplementary Note 20: Narrowband filtered inverse FFT and short time FFT (STFT) analysis:

Figure 46. Time-domain reconstruction of vibrational wave-packet oscillations by inverse Fourier filtering. **a.** CW-Resonant Raman spectra of the Ref-P, TS-P2, TS-P3 at the high frequency regime (excitation wavelength, 785 nm). **b.** Integrated Fourier transform IVS spectra for the probe wavelength range, $\lambda = 890\text{--}920\text{ nm}$ which corresponds to the polymer backbone chromophore $S_1 \rightarrow S_n$ photo-induced absorption **c,d.** The transient absorption signal in the $890\text{--}930\text{ nm}$ probe range was first detrended using a 200 fs moving-average window to isolate coherent residuals for TS-P3(**c**) and Ref-P(**d**). The residual trace was then narrow-band filtered in the frequency domain within two wavenumber windows ($1400\text{--}1450\text{ cm}^{-1}$ and $1500\text{--}1550\text{ cm}^{-1}$) by zeroing all other spectral components and performing an inverse fast Fourier transform (iFFT). This procedure retrieves the oscillatory components associated with individual vibrational modes, while the analytic-signal (Hilbert transform) envelope provides their temporal dephasing behaviour. The faster decay of the $1500\text{--}1550\text{ cm}^{-1}$ component compared with the $1400\text{--}1450\text{ cm}^{-1}$ band indicates a shorter coherence lifetime of the mode driving electron transfer in TS-P3.

We analysed the photoinduced absorption region ($890\text{--}920\text{ nm}$) corresponding to the $S_1 \rightarrow S_n$ transition in the polymer, as presented in the main text. As shown in Figure 3d (main figure) and reproduced in Supplementary Figure 46a, a significant reduction in the FFT amplitude of the 1507 cm^{-1} mode is observed for TS-P3 compared with TS-P2 and Ref-P. In contrast, the CW-resonant Raman spectra of all three systems exhibit nearly identical

vibrational features (Figure 3e, reproduced as Supplementary Figure 46b). This indicates that the loss of intensity in the 1507 cm^{-1} mode for TS-P3 originates from vibrational decoherence processes occurring within the excited-state manifold, consistent with previous observations in small-molecule systems in solution^{22,57}.

To further examine the time-domain behaviour of these modes, we applied a narrow-band frequency mask and retrieved the corresponding temporal profiles via inverse FFT, following the procedure described in Ref⁵⁸. The resulting data are shown in Supplementary Figures 46c (TS-P3) and 46d (Ref-P). The dephasing times extracted for TS-P3 are 1182 fs in the $1400\text{--}1450\text{ cm}^{-1}$ region and 870 fs in the $1500\text{--}1550\text{ cm}^{-1}$ region, while for Ref-P the corresponding values are 1139 fs and 1041 fs, respectively. These results demonstrate that the 1512 cm^{-1} mode in TS-P3 exhibits a faster dephasing rate than in Ref-P, whereas the 1428 cm^{-1} spectator mode shows comparable dephasing times in both systems. This is consistent with the quantum chemical calculations reported in Fig. 4 of the main text. The accelerated decoherence of the 1512 cm^{-1} mode in TS-P3 supports its role as a driving mode in the electron-transfer process. Such rapid dephasing of a driving vibrational mode is often attributed to nuclear nesting effects. We further note that this mode likely possesses an even faster component, as the electron transfer occurs on a sub-15 fs timescale; however, the early-time dynamics are obscured by coherent artefacts and cross-phase modulation, which introduce high-frequency oscillatory contributions.

We performed a narrow-band time-domain analysis and a short-time Fourier transform (STFT⁵⁸) following the approach in ref⁵⁸, using a 200 fs window with 20 fs overlap. The STFT spectrogram (Supplementary Fig. 47) shows a clear temporal lag in the appearance of the 1283 cm^{-1} mode, confirming that it arises from charge-transfer dynamics rather than direct impulsive excitation. Complementary inverse FFT analysis of the $1400\text{--}1450\text{ cm}^{-1}$ and $1500\text{--}1550\text{ cm}^{-1}$ windows reveals faster dephasing of the 1512 cm^{-1} DPP-localised mode in TS-P3 ($\tau \approx 870\text{ fs}$) relative to Ref-P ($\tau \approx 1040\text{ fs}$), while the spectator 1428 cm^{-1} mode remains unchanged. These results reveal the distinct temporal evolution of vibrational coherences, showing the 1283 cm^{-1} mode as a product-state vibration generated upon charge transfer, and the analysis further support our assignment of the 1512 cm^{-1} mode as the driving coordinate for the ultrafast electron-transfer process.

Figure 47. Short-time FT (STFT) analysis of vibrational dynamics in TS-P3 at 890–930 nm probe window. (a) Transient absorption time trace (red) with its 200 fs moving-average background (blue) and the extracted residual oscillations (orange). The 200 fs moving average method is used for subtracting the background to remove the low-frequency component from the time domain data. The residual isolates coherent vibrational motion superimposed on the electronic response. (b) Fast Fourier transform (FFT) of the residual signal showing distinct vibrational modes between 200 and 1800 cm^{-1} . (c) Short-time Fourier transform (STFT) spectrogram revealing the temporal evolution of these modes. The dashed line marks the 1283 cm^{-1} feature, characteristic of the mode activated by electron-transfer dynamics. The colour scale represents oscillation power (a.u.).

Minor issues:

-The abstract is very vague and should refer to the real types of systems being discussed in the text.

Response: We thank the reviewer for this valuable comment. The abstract has been revised to be more specific and representative of the studied systems:

- Specified the system as *through-space polymer-non-fullerene (PDI-BDT-DPP) heterojunctions*.

- Added details on *precisely controlled geometry, weak donor–acceptor coupling, and <100 meV energy offset.*
- Identified a *25 fs high-frequency vibrational mode on the PDI acceptor’s potential energy surface.*
- Indicated that this *DPP-centred mode drives ultrafast charge transfer* through vibronic coupling.

We hope these revisions make the abstract clearer, more informative, and directly aligned with the content of the manuscript.

Action:

Revised abstract:

Excited-state charge transfer (CT) plays a crucial role in organic photovoltaics, photocatalysis, and photodetection applications. Traditionally, rapid CT is thought to require a large energy offset and strong electronic coupling between donor and acceptor molecules —conditions that maybe detrimental to overall device performance. The role of vibrations in driving CT and how this might ultimately govern rates of charge transfer in organic heterojunctions is still an open question. Here, we investigate through-space polymer–non-fullerene model heterojunctions in which a perylene diimide (PDI) acceptor is covalently tethered to a low-bandgap push–pull (BDT-DPP) polymer donor. These systems feature precisely controlled intermolecular geometry and an exceptionally small energy offset (<100 meV) between frontier orbitals, with weak donor–acceptor coupling in the Franck–Condon region. Despite the low-energy offset, we achieve a CT timescale less than 15 fs, which is accompanied via the launch of coherent wavepackets along the coordinate of a high frequency mode of period 25fs, on the PDI acceptor’s potential energy surface. Using evidence from steady-state and ultrafast electronic and vibrational spectroscopy, combined with electronic structure calculations we identify the specific DPP-centered driving modes that enable such unprecedented CT rates in purely organic systems, by mixing the singlet Frenkel exciton state and CT state potential energy surfaces following photoexcitation. Our results show that it is possible to achieve very

rapid CT, limited ultimately by the vibrational time-period of high-frequency modes, despite the lack of large energy offsets or strong coupling between donor and PDI acceptor units in the ground state of the system. This has important implications for the design of organic materials for organic photovoltaic, photocatalysis and optoelectronics.

-Figure 2a: "S₀→S_n" absorption is presumably "S₁→S_n" absorption as stated on page 7.

Response: We thank the reviewer for pointing out this mistake and apologise for the oversight.

Action: The label in Figure 2a has been corrected from "S₀ → S_n" to "S₁ → S_n" in the revised manuscript.

-Figure 2b: What is the origin of the relatively slow rise in the Ref-P signal? This is also present to a lesser extent in the TS-P2 data.

Response: We thank the reviewer for this comment. We assign the slow sub picosecond rise in the Ref-P signal to the dielectric relaxation or excited state solvation. The assignment is supported by the push-pull dyes (monomeric analogues of the Ref-P) as seen before (ref⁵⁹).

Action:

Revised supplementary information excerpt (Supplementary Note 17)

Such near-instantaneous transfer is consistent with the vibronically coherent regime discussed in the main text and is further supported by the 100 fs-resolved transient absorption dynamics (Fig. 2b, main text), where slower sub-picosecond solvation relaxation is evident in Ref-P and TS-P2 but absent in TS-P3. Together, these findings

confirm that the < 15 fs component represents the coherent electron transfer event rather than subsequent solvent or structural relaxation.

-Figure 2d-f: It would be helpful to show a "zero-line" to guide the eye for when the transient absorption signals cross change sign.

Response: We thank the reviewer for this helpful suggestion.

Action: A zero-line has been added to Figures 2d-f in the revised manuscript to guide the eye when the transient absorption signals cross through zero.

-Figure 3b: what is the range of integration for the Fourier transform data?

Response: We thank the reviewer for this question and agree that the integration window should be stated explicitly.

Action: We have clarified in the revised manuscript that the Fourier transform in Figure 3b was performed over the range 40–1700 fs, and this information has been added to the main text figure caption.

-Page 9, line 214: "TP-P3" is presumably "TS-P3"

Response: We thank the reviewer for pointing out the typo.

Action: We have corrected the 'TP-P3' to 'TS-P3'.

-Page 10/Supplementary Note 5: *The agreement of the transient absorption spectrum with the spectrum of the chemically reduced PDI is not great. What is the origin of this disagreement? Aggregation is known to cause such broadening and shifting of anion spectra. Its use as supporting evidence is undermined by the lack of agreement and discussion.*

Response: We thank the reviewer for highlighting this important point. We agree that the spectral differences between the transient absorption (TA) features and the chemically reduced PDI⁻ reference require further clarification. As reported by other groups⁶⁰, aggregation of PDI⁻ anions typically leads to pronounced spectral broadening accompanied by a 200–300 meV blue shift. In contrast, the chemically reduced PDI in our study shows peaks at 714 nm and 797 nm, whereas the transient PDI⁻ features observed in TS-P3 appear at 734 nm and 825 nm—corresponding to moderate red-shifts of \approx 46 meV and 52 meV, respectively, together with some broadening.

The opposite direction of this shift indicates that aggregation is not the dominant origin of the discrepancy. Instead, the red-shift most likely arises from local environmental

effects within the polymer–PDI heterojunction—specifically, electrostatic stabilization of the PDI anion by the adjacent polymer hole (reflecting differences in counter-ion distribution and solvation) and dielectric confinement in the nanoscale mixed phase. Additionally, static conformational disorder within the polymer matrix may introduce a distribution of effective electron–hole distances, contributing to the observed broadening of the PDI anion band in the TA spectra.

Importantly, following the reviewer’s helpful suggestion, we have now reassessed the comparison used to discuss the 1283 cm^{-1} mode. We directly compare the spectral distribution of this vibrational coherence with the PDI^- feature retrieved from the transient absorption data, which shows excellent correspondence in both position and shape (Supplementary Fig. 6b). The chemically reduced PDI^- spectrum is now retained only as a qualitative reference. We thank the reviewer for prompting this improvement, which has substantially strengthened the discussion in Supplementary Note 5 and clarified the excited-state origin of the 1283 cm^{-1} mode.

Action:

Main manuscript:

“In contrast, the vibrational coherence at 1283 cm^{-1} (ω^β) obtained by resonant impulsive excitation of the polymer backbone in TS-P3 shows its strongest amplitude at probe wavelengths between 650-850 nm (black spectra, Figure 3c, top). This maximum amplitude closely resembles the absorption of the PDI radical anion (orange spectra, Figure 3c, top).”

The section above is replaced by-

“In contrast, the vibrational coherence at 1283 cm^{-1} (ω^β), obtained by resonant impulsive excitation of the polymer backbone in TS-P3, shows its strongest amplitude at probe wavelengths between 650 and 850 nm (black trace, Fig. 3c, top). This spectral envelope closely matches the PDI radical-anion feature retrieved from the transient absorption data (red trace, Fig. 3c, top), and is further qualitatively complemented by the absorption profile of the chemically reduced PDI^- species (blue trace, Fig. 3c, top).”

Subfig. 3c (revised) (Top) The black line represents the FT amplitude for the vibrational coherence at 1283 cm⁻¹ mode as a function of probe wavelength while resonantly exciting TS-P3 with the 'red impulsive pump'. The orange filled blue dotted line area represents the absorbance of PDI radical anion and the red line corresponds to the PDI anion absorption retrieved from TA (supplementary note XX). (bottom) the FT amplitude for the vibrational coherence at 1295 cm⁻¹ mode as a function of probe wavelength while off-resonantly exciting PDI with the 'red impulsive pump'.

Supplementary Note 5:

In Supplementary Fig. 6, the differential transmission signal ($\Delta T/T$) for the TS-P3 system is presented at 20 fs and 40 fs, represented by the solid dark blue and green curves, respectively. As noted in Fig. 2d (main text), the Ref-P system exhibits no dynamics on this ultrafast timescale. Therefore, the difference $\Delta(\Delta T/T) = \Delta T/T(20 \text{ fs}) - \Delta T/T(40 \text{ fs})$ should predominantly reflect the absorption of the PDI radical anion., provided there is minimal spectral contribution from other features, such as the hole polaron associated with the polymer backbone. It is important to note that the PDI radical anion spectrum qualitatively retrieved from the transient absorption (TA) data (red curve, Supplementary Fig. 6) is both broadened and red-shifted relative to the chemically reduced PDI reference (blue shaded area, Supplementary Fig. 6a–b). The chemically reduced PDI exhibits absorption peaks at 714 nm and 797 nm, whereas the transiently generated PDI⁻ features in TS-P3 appear at 734 nm and 825 nm—corresponding to moderate red-shifts of approximately 46 meV and 52 meV, respectively, accompanied by noticeable broadening. This red-shift likely originates from several factors. First, the chemical structure differs: in TS-P3 the PDI is covalently linked to the polymer backbone via methoxy-dimethylamine or 1,3-dimethoxypropane tethers (Supplementary Fig. 6c), which can modify the frontier orbital energies compared with the isolated PDI-OMe reference. Second, the charge-stabilizing environment is distinct. In the chemically reduced sample, the PDI anion is generated in a polar solvent using tetrakis(diethylamino)ethylene, where the counter-ions are fully solvated and spatially separated. In contrast, within TS-P3 the electron transfer produces a tightly bound

charge-transfer (CT) state in which the hole polaron on the polymer backbone acts as a proximal counter-charge. The resulting local electrostatic stabilization, partial solvation, and reduced dielectric screening can shift the PDI⁻ absorption to lower energy.

Finally, the broader spectral profile observed in the TA-retrieved spectrum can be attributed to static conformational disorder within the polymer matrix, which introduces a distribution of effective electron-hole separations and hence a spread in local stabilization energies. Collectively, these effects explain the moderate red-shift and broadening of the transient PDI⁻ spectrum relative to the chemically reduced reference.

As the chemically reduced PDI⁻ spectrum serves only as a qualitative reference, we now directly compare the spectral distribution of the 1283 cm⁻¹ vibrational coherence (black curve, Supplementary Fig. 6b) with the PDI⁻ absorption feature retrieved from the transient absorption data (red curve). This comparison reveals a markedly better correspondence, both in spectral position and in overall envelope shape, confirming that the coherence is resonantly detected through the transiently generated PDI radical anion rather than through the ground-state PDI. Notably, the nodal behaviour near the peak of the PDI⁻ absorption band arises from a phase inversion of the vibrational coherence signal, as expected for modulation of the excited-state charge-transfer absorption. This agreement provides strong evidence that the 1283 cm⁻¹ mode originates from the PDI radical anion formed upon ultrafast charge transfer.

Supplementary Fig. 6| Spectral characterization of the PDI radical anion: (a) The blue shaded area with dotted line corresponds to the absorption spectra of chemically reduced PDI radical anion. (b) The PDI anion feature derived from TA and from chemical reduction is overlaid with the spectral intensity profile of the 1283 cm⁻¹ mode (c) The PDI structure that is used for the chemical reduction.

-Page 21, the figure call in the methodology section is incomplete.

Response: We thank the reviewer for pointing out the error in citing the SHG-FROG characterization in the methodology section.

Action: The figure call has been corrected from “Supplementary Fig. xx12” to “Supplementary Fig. 5” on Page 21 of the methodology section in the revised manuscript.

References

1. Royakkers, J. *et al.* Synthesis of model heterojunction interfaces reveals molecular-configuration-dependent photoinduced charge transfer. *Nat Chem* **16**, 1453–1461 (2024).
2. Winte, K. *et al.* Vibronic coupling-driven symmetry breaking and solvation in the photoexcited dynamics of quadrupolar dyes. *Nat Chem* <https://doi.org/10.1038/s41557-025-01908-7> (2025) doi:10.1038/s41557-025-01908-7.
3. Karuthedath, S. *et al.* Intrinsic efficiency limits in low-bandgap non-fullerene acceptor organic solar cells. *Nat Mater* **20**, 378–384 (2021).
4. Karki, A. *et al.* Unifying Charge Generation, Recombination, and Extraction in Low-Offset Non-Fullerene Acceptor Organic Solar Cells. *Adv Energy Mater* **10**, (2020).
5. Liu, Y., Zuo, L., Shi, X., Jen, A. K. Y. & Ginger, D. S. Unexpectedly Slow Yet Efficient Picosecond to Nanosecond Photoinduced Hole-Transfer Occurs in a Polymer/Nonfullerene Acceptor Organic Photovoltaic Blend. *ACS Energy Lett* **3**, 2396–2403 (2018).
6. Zhou, G. *et al.* Marcus Hole Transfer Governs Charge Generation and Device Operation in Nonfullerene Organic Solar Cells. *ACS Energy Lett* **6**, 2971–2981 (2021).
7. Mayerhöffer, U., Gsänger, M., Stolte, M., Fimmel, B. & Würthner, F. Synthesis and molecular properties of acceptor-substituted squaraine dyes. *Chemistry - A European Journal* **19**, 218–232 (2013).
8. Ehara, T. *et al.* Dynamic Excited-State Localization Induced by Jahn-Teller Distortion Observed by Coherent Vibrational Spectroscopy. *J Am Chem Soc* <https://doi.org/10.1021/jacs.5c06020> (2025) doi:10.1021/jacs.5c06020.

9. Kuramochi, H. *et al.* Femtosecond Polarization Switching in the Crystal of a [CrCo] Dinuclear Complex. *Angewandte Chemie - International Edition* **59**, 15865–15869 (2020).
10. Yoneda, Y., Sotome, H., Mathew, R., Lakshmana, Y. A. & Miyasaka, H. Non-condon Effect on Ultrafast Excited-State Intramolecular Proton Transfer. *Journal of Physical Chemistry A* **124**, 265–271 (2020).
11. Zewail, A. H. Optical molecular dephasing: principles of and probings by coherent laser spectroscopy. *Acc Chem Res* **13**, 360–368 (1980).
12. Ghosh, P. *et al.* Decoupling excitons from high-frequency vibrations in organic molecules. *Nature* **629**, 355–362 (2024).
13. Ostroverkhova, O. Organic Optoelectronic Materials: Mechanisms and Applications. *Chemical Reviews* vol. 116 13279–13412 Preprint at <https://doi.org/10.1021/acs.chemrev.6b00127> (2016).
14. Meskers, S. C. J., Janssen, R. A. J., Saes, B. W. H., Lutz, M. & Wienk, M. M. Tuning the optical characteristics of diketopyrrolopyrrole molecules in the solid state by alkyl side chains. *Journal of Physical Chemistry C* **124**, 25229–25238 (2020).
15. Yang, J. *et al.* Isoindigo-Based Polymers with Small Effective Masses for High-Mobility Ambipolar Field-Effect Transistors. *Advanced Materials* **29**, (2017).
16. Liang, S. *et al.* Double-Cable Conjugated Polymers with Pendent Near-Infrared Electron Acceptors for Single-Component Organic Solar Cells. *Angewandte Chemie - International Edition* **61**, (2022).
17. Zhang, F. *et al.* Soluble polythiophenes with pendant fullerene groups as double cable materials for photodiodes. *Advanced Materials* **13**, 1871–1874 (2001).
18. Tan, ao *et al.* Synthesis and Photovoltaic Properties of a Donor-Acceptor Double-Cable Polythiophene with High Content of C 60 Pendant. <https://doi.org/10.1021/ma070052> (2007) doi:10.1021/ma070052.
19. Feng, G. *et al.* Thermal-Driven Phase Separation of Double-Cable Polymers Enables Efficient Single-Component Organic Solar Cells. *Joule* **3**, 1765–1781 (2019).
20. Wu, Y. *et al.* A conjugated donor-acceptor block copolymer enables over 11% efficiency for single-component polymer solar cells. *Joule* **5**, 1800–1815 (2021).
21. He, Y. *et al.* Industrial viability of single-component organic solar cells. *Joule* vol. 6 1160–1171 Preprint at <https://doi.org/10.1016/j.joule.2022.05.008> (2022).
22. Rafiq, S., Fu, B., Kudisch, B. & Scholes, G. D. Interplay of vibrational wavepackets during an ultrafast electron transfer reaction. *Nat Chem* **13**, 70–76 (2021).

23. Kong, J. *et al.* Robust vibrational coherence protected by a core-shell structure in silver nanoclusters. *Chem Sci* **15**, 6906–6915 (2024).
24. Liebel, M., Schnedermann, C., Wende, T. & Kukura, P. Principles and Applications of Broadband Impulsive Vibrational Spectroscopy. *Journal of Physical Chemistry A* **119**, 9506–9517 (2015).
25. Worth, G. A., Beck, H., Jäckle, A. & Meyer, H.-D. Quantics: A general purpose package for Quantum molecular dynamics simulations ☆ The multiconfiguration time-dependent Hartree method: A highly efficient algorithm for propagating wavepackets. *Comput Phys Commun* **248**, 107040 (2020).
26. De Sio, A. *et al.* Intermolecular conical intersections in molecular aggregates. *Nat Nanotechnol* **16**, 63–68 (2021).
27. Popp, W., Brey, D., Binder, R. & Burghardt, I. Quantum Dynamics of Exciton Transport and Dissociation in Multichromophoric Systems. *Annu Rev Phys Chem* **72**, 591–616 (2021).
28. Aarabi, M. *et al.* Quantum-Classical Protocol for Efficient Characterization of Absorption Lineshape and Fluorescence Quenching upon Aggregation: The Case of Zinc Phthalocyanine Dyes. *J Chem Theory Comput* **19**, 5938–5957 (2023).
29. Meyer, H. D. Studying molecular quantum dynamics with the multiconfiguration time-dependent Hartree method. *Wiley Interdiscip Rev Comput Mol Sci* **2**, 351–374 (2012).
30. Beck, M. The multiconfiguration time-dependent Hartree (MCTDH) method: a highly efficient algorithm for propagating wavepackets. *Phys Rep* **324**, 1–105 (2000).
31. Vendrell, O. & Meyer, H. D. Multilayer multiconfiguration time-dependent Hartree method: Implementation and applications to a Henon-Heiles Hamiltonian and to pyrazine. *Journal of Chemical Physics* **134**, (2011).
32. Meyer, H. D. & Worth, G. A. Quantum molecular dynamics: Propagating wavepackets and density operators using the multiconfiguration time-dependent Hartree method. *Theor Chem Acc* **109**, 251–267 (2003).
33. Manthe, U. A multilayer multiconfigurational time-dependent Hartree approach for quantum dynamics on general potential energy surfaces. *J Chem Phys* **128**, (2008).
34. Popp, W., Brey, D., Binder, R. & Burghardt, I. Quantum Dynamics of Exciton Transport and Dissociation in Multichromophoric Systems. *Annu Rev Phys Chem* **72**, 591–616 (2021).

35. Segalina, A. *et al.* How the Interplay among Conformational Disorder, Solvation, Local, and Charge-Transfer Excitations Affects the Absorption Spectrum and Photoinduced Dynamics of Perylene Diimide Dimers: A Molecular Dynamics/Quantum Vibronic Approach. *J Chem Theory Comput* **18**, 3718–3736 (2022).
36. Troisi, A. Charge transport in high mobility molecular semiconductors: classical models and new theories. *Chem Soc Rev* **40**, 2347 (2011).
37. Hsu, C.-P., You, Z.-Q. & Chen, H.-C. Characterization of the Short-Range Couplings in Excitation Energy Transfer. *The Journal of Physical Chemistry C* **112**, 1204–1212 (2008).
38. Voityuk, A. A. Estimation of electronic coupling in π -stacked donor-bridge-acceptor systems: Correction of the two-state model. *Journal of Chemical Physics* **124**, 1–7 (2006).
39. Yang, C.-H. & Hsu, C.-P. A multi-state fragment charge difference approach for diabatic states in electron transfer: Extension and automation. *J Chem Phys* **139**, 154104 (2013).
40. Cupellini, L., Corbella, M., Mennucci, B. & Curutchet, C. Electronic energy transfer in biomacromolecules. *WIREs Computational Molecular Science* **9**, 1–23 (2019).
41. Nottoli, M. *et al.* The role of charge-transfer states in the spectral tuning of antenna complexes of purple bacteria. *Photosynth Res* **137**, 215–226 (2018).
42. Tölle, J., Cupellini, L., Mennucci, B. & Neugebauer, J. Electronic couplings for photo-induced processes from subsystem time-dependent density-functional theory: The role of the diabatization. *Journal of Chemical Physics* **153**, 184113 (2020).
43. Giannini, S. *et al.* Exciton transport in molecular organic semiconductors boosted by transient quantum delocalization. *Nat Commun* **13**, 2755 (2022).
44. Hong, Y., Schlosser, F., Kim, W., Würthner, F. & Kim, D. Ultrafast Symmetry-Breaking Charge Separation in a Perylene Bisimide Dimer Enabled by Vibronic Coupling and Breakdown of Adiabaticity. *J Am Chem Soc* **144**, 15539–15548 (2022).
45. Lin, C., Kim, T., Schultz, J. D., Young, R. M. & Wasielewski, M. R. Accelerating symmetry-breaking charge separation in a perylenediimide trimer through a vibronically coherent dimer intermediate. *Nat Chem* **14**, 786–793 (2022).

46. Bartynski, A. N. *et al.* Symmetry-breaking charge transfer in a zinc chlorodipyrin acceptor for high open circuit voltage organic photovoltaics. *J Am Chem Soc* **137**, 5397–5405 (2015).
47. Avila Ferrer, F. J. & Santoro, F. Comparison of vertical and adiabatic harmonic approaches for the calculation of the vibrational structure of electronic spectra. *Physical Chemistry Chemical Physics* **14**, 13549 (2012).
48. Cerezo, J. & Santoro, F. FCclasses3 : Vibrationally-resolved spectra simulated at the edge of the harmonic approximation. *J Comput Chem* **44**, 626–643 (2023).
49. Aarabi, M. *et al.* Quantum-Classical Protocol for Efficient Characterization of Absorption Lineshape and Fluorescence Quenching upon Aggregation: The Case of Zinc Phthalocyanine Dyes. *J Chem Theory Comput* <https://doi.org/10.1021/acs.jctc.3c00446> (2023) doi:10.1021/acs.jctc.3c00446.
50. Worth, G. A. Quantics: A general purpose package for Quantum molecular dynamics simulations. *Comput Phys Commun* **248**, 107040 (2020).
51. Kippelen, B. & Brédas, J. L. Organic photovoltaics. *Energy and Environmental Science* vol. 2 251–261 Preprint at <https://doi.org/10.1039/b812502n> (2009).
52. Janssen, R. A. J. & Nelson, J. Factors limiting device efficiency in organic photovoltaics. *Advanced Materials* vol. 25 1847–1858 Preprint at <https://doi.org/10.1002/adma.201202873> (2013).
53. Eastham, N. D. *et al.* Hole-Transfer Dependence on Blend Morphology and Energy Level Alignment in Polymer: ITIC Photovoltaic Materials. *Advanced Materials* **30**, (2018).
54. Zheng, Z. *et al.* Efficient Charge Transfer and Fine-Tuned Energy Level Alignment in a THF-Processed Fullerene-Free Organic Solar Cell with 11.3% Efficiency. *Advanced Materials* **29**, (2017).
55. Menke, S. M., Luhman, W. A. & Holmes, R. J. Tailored exciton diffusion in organic photovoltaic cells for enhanced power conversion efficiency. *Nat Mater* **12**, 152–157 (2013).
56. Yuan, J. *et al.* Single-Junction Organic Solar Cell with over 15% Efficiency Using Fused-Ring Acceptor with Electron-Deficient Core. *Joule* **3**, 1140–1151 (2019).
57. Yoneda, Y. *et al.* Vibrational Dephasing along the Reaction Coordinate of an Electron Transfer Reaction. *J Am Chem Soc* **143**, 14511–14522 (2021).
58. Hong, Y., Schlosser, F., Kim, W., Würthner, F. & Kim, D. Ultrafast Symmetry-Breaking Charge Separation in a Perylene Bisimide Dimer Enabled by Vibronic

Coupling and Breakdown of Adiabaticity. *J Am Chem Soc* **144**, 15539–15548 (2022).

59. Liu, Z. *et al.* Intersystem Crossing in Acceptor-Donor-Acceptor Type Organic Photovoltaic Molecules Promoted by Symmetry Breaking in Polar Environments. *Journal of Physical Chemistry Letters* **13**, 10305–10311 (2022).
60. Li, H. & Wenger, O. S. Photophysics of Perylene Diimide Dianions and Their Application in Photoredox Catalysis. *Angewandte Chemie - International Edition* **61**, (2022).

Response to the reviewers' comments

Reviewer #1 (Remarks to the Author):

After the first round of reviewing, the manuscript has been improved substantially. The authors have responded to the comments and suggestions of both reviewers. They have added new measurements and new data analysis. This is very helpful and much appreciated.

In particular, I have enjoyed seeing the excited state quantum dynamics simulations which are now added as a new section in the Supplementary Information. All in all, the extensive revisions made by the reviewers have supported the case of the authors quite strongly. I think that the manuscript is now much closer to being accepted for publication in Nature Communications.

I want to ask the authors to fully address the following points before recommending publication of this article.

Response: We thank the reviewer for their careful re-evaluation of the manuscript and for their thoughtful and positive assessment. We are pleased that the reviewer considers the manuscript to be substantially improved following revision, and that the additional measurements, data analysis, and the newly added excited-state quantum dynamics simulations in the Supplementary Information are regarded as valuable and informative. We appreciate the reviewer's recognition that these extensive revisions have strengthened the overall case for the work.

We address each of the remaining points raised by the reviewer in detail below.

a) I very much appreciate the details that are now given about the data analysis (SFig 41). I must say that I have my doubts about the claim ET time of 14.9 +/- 1 fs. I understand that the subtraction of coherent artefacts is challenging and U see that this has been done carefully. Yet, I think that red curve in 41b will also describe the data if an ET time of 20 or 25 fs is assumed (depending on how one takes into the account remaining "artefacts". I

think that it is fair that the ET is fast (< 25 fs or so) but I do not think that the ET time can be extracted with one fs precision. I request to change this.

Response: We thank the reviewer for raising this important point. We agree that the extracted ultrafast electron-transfer timescale can be sensitive to the temporal window used in the analysis. As shown in the figure, when different upper time limits are considered, the fast time constant is distributed between approximately 12.8 fs and 20.1 fs.

This variation arises because, following the rapid electron-transfer event, a weak slower signal component is present, which we attribute to dielectric relaxation of the newly formed charged species. To account for this effect more rigorously, we have extended the fitting window to 2000 fs. After careful subtraction of the coherent artefact, the kinetics were analysed using a bi-exponential model. This yields a fast component of 18.1 ± 3.1 fs, with a conservative upper bound of 21.2 fs, fully consistent with the reviewer's suggestion that the electron transfer is ultrafast (< 25 fs) but cannot be determined with one-femtosecond precision. Importantly, this refinement does not alter the physical interpretation of the results: the electron-transfer process remains faster than the period of the coherently generated vibrational mode (~ 26 fs), and therefore still occurs within the vibrationally coherent regime discussed in the manuscript. We have revised the relevant sections of the main text and Supplementary Information accordingly and now report the bi-exponential analysis throughout.

Action:

SI excerpt

The resulting differential kinetics (Supplementary Fig. 41b) were fit with a mono-exponential decay function, yielding a time constant of $\tau = 14.9 \pm 1$ fs. This value represents the experimentally resolvable upper limit of the CT timescale, approaching the 11 fs instrument response function. The resulting differential kinetics (Supplementary Fig. 41b) were analysed using a biexponential model comprising a dominant ultrafast component and a much weaker slower contribution. The fast component is attributed to electron transfer from the polymer backbone to the PDI acceptor, while the slower component likely reflects secondary processes following charge separation, such as dielectric or structural relaxation of the newly formed charge-separated state. This analysis yields a fast characteristic timescale of 18.1 ± 3.1 fs, corresponding to a conservative upper bound of 21.2 fs for the electron-transfer process. Importantly, this analysis supports the physical interpretation of the results: the electron-transfer process is faster than the period of the coherently generated vibrational mode (~ 26 fs), confirming that charge transfer occurs on a sub-vibrational timescale.

Supplementary Fig. 41 | b, Difference trace (TS-P3 – Ref) fitted with a dominant ultrafast decay component (red line), corresponding to an electron-transfer timescale of ~ 18 fs (upper bound ~ 21 fs).

Main excerpt

All mentions of the electron-transfer timescale in the main text have been revised for consistency with the updated analysis. For example:

... which yielded a fast component of 18.1 ± 3.1 fs, corresponding to a conservative upper bound of ~ 21.2 fs for the electron-transfer process ~~a mono-exponential decay with $\tau = 14.9 \pm 1$ fs, establishing an upper limit for the electron transfer process~~

b) I guess that the time axis in SI Fig. 4 should be “ps” – not “fs” ;-)

Response: We thank the reviewer for identifying this error. The time axis in Supplementary Fig. 4 was incorrectly labelled; the correct unit is ps rather than fs.

Action: This has now been corrected in the revised Supplementary Information.

c) The title in SFig. 49 should probably read “with/without linkers”.

Response: We thank the reviewer for noting this typographical error.

Action: The title of Supplementary Fig. 49 has been corrected to read “with/without linkers.”

d) I acknowledge the discussion about Herzberg Teller couplings and agree qualitatively with what is said. Yet, I think that a much more careful analysis of the (possibly short-lived) excited state vibrations is needed before conclusions can be drawn. I would therefore suggest not to make too many strong statements about the origin of these vibrational coherences.

Response: We thank the reviewer for this important point and agree that caution is required when interpreting the origin of short-lived excited-state vibrational coherences. In the revised manuscript, we no longer make definitive statements assigning the observed coherences to a specific Herzberg–Teller coupling mechanism. Instead, the discussion is framed more conservatively, emphasising that the observations are consistent with vibronic coupling without uniquely establishing a single microscopic origin.

In addition, we have included further time–frequency analyses (narrowband-filtered inverse FFT and short-time Fourier transform) in the revised Supplementary Information (Supplementary Note 20) to better characterise the vibrational coherence, while explicitly noting the limitations of such methods at the earliest time delays. Those analysis further supported the mechanism that is proposed. As suggested by the

reviewer, we have moderated the strength of several mechanistic statements in the main text. We hope that these revisions adequately address the reviewer's concern.

Action: Relevant changes in the main text are highlighted in red.

Since the authors have revised the manuscript very thoroughly, I gladly recommend publication of this article after these minor points have been carefully addressed.

We hope that the remaining minor points raised by the reviewer have now been fully addressed, and we sincerely thank the reviewer for their careful reading and constructive feedback.

Reviewer #2 (Remarks to the Author):

The authors have made significant efforts to address the reviewer comments, and the manuscript is much improved as a result. The manuscript is suitable for publication in Nature Communications.

Response: We thank the reviewer for their careful re-evaluation of the manuscript and for their positive assessment. We are pleased that the revisions have addressed the reviewer's concerns and that the manuscript has been substantially improved as a result. We appreciate the reviewer's time and constructive feedback, which have strengthened the clarity and quality of the work.